# Non-competitive resource exploitation within mosquito shapes within-host malaria infectivity and virulence

G. Costa [ID] [1], M. Gildenhard[1], M. Eldering[1,2], R.L. Lindquist [ID] [3], A.E. Hauser[3,4], R. Sauerwein[2], C. Goosmann[5], V. Brinkmann [ID] [5], P. Carrillo-Bustamante [ID] [1] & E.A. Levashina[1]

Malaria is a fatal human parasitic disease transmitted by a mosquito vector. Although the evolution of within-host malaria virulence has been the focus of many theoretical and empirical studies, the vector's contribution to this process is not well understood. Here, we explore how within-vector resource exploitation would impact the evolution of within-host *Plasmodium* virulence. By combining within-vector dynamics and malaria epidemiology, we develop a mathematical model, which predicts that non-competitive parasitic resource exploitation within-vector restricts within-host parasite virulence. To validate our model, we experimentally manipulate mosquito lipid trafficking and gauge within-vector parasite development and within-host infectivity and virulence. We find that mosquito-derived lipids determine within-host parasite virulence by shaping development (quantity) and metabolic activity (quality) of transmissible sporozoites. Our findings uncover the potential impact of within-vector environment and vector control strategies on the evolution of malaria virulence.

[1] Vector Biology Unit, Max Planck Institute for Infection Biology (MPIIB), 10117 Berlin, Germany. [2] Department of Medical Microbiology, Radboud University Medical Center, PO Box 91016500 HB Nijmegen, The Netherlands. [3] Immunodynamics, German Rheumatism Research Centre (DRFZ), 10117 Berlin, Germany. [4] Immune Dynamics and Intravital Microscopy, Charité-Universitätsmedizin, 10117 Berlin, Germany. [5] Microscopy Core Facility, Max Planck Institute for Infection Biology (MPIIB), 10117 Berlin, Germany. These authors contributed equally: M. Gildenhard, M. Eldering. Correspondence and requests for materials should be addressed to E.A.L. (email: levashina@mpiib-berlin.mpg.de)

Malaria is caused by the vector-borne protozoan parasite *Plasmodium falciparum* and kills 429,000 people annually, predominantly in sub-Saharan Africa[1]. The unacceptably high malaria mortality is tightly linked to the parasite's capacity to cause harm to the human host (here defined as within-host parasite virulence). Therefore, predicting the evolution of *Plasmodium* virulence has important epidemiological and socio-medical implications for designing malaria control strategies. The life cycle of malaria parasites entails two hosts: an intermediate mammalian host (hereafter referred to as host) and a definitive mosquito host (hereafter referred to as a vector). Theoretical and empirical studies have focused on regulation of within-host malaria virulence, including contributions of parasite genetic factors and host immune responses, and on the link between virulence and host-to-vector transmission[2–4]. Further reports showed that genetically-encoded within-host parasite virulence increases the production of infective-to-vector forms and, thereby, parasite transmission to the vector[5]. However, the vector's contribution to within-host virulence remains poorly understood.

Females anopheline mosquitoes feed on blood to initiate ovary development. Therefore, only females ingest parasites and contribute to malaria transmission. During a blood meal, the fusion of blood-borne sexual forms of parasites engenders motile ookinetes that within 2 days after infection traverse the midgut epithelium and reach the basal side of the midgut to round-up into oocysts. In the next 10–15 days, oocysts undergo sporogony, a process that generates thousands of infective-to-human sporozoites. Sporozoites egress from the oocysts and migrate to the mosquito salivary glands to reach full maturity and infectivity[6]. Nutrients taken up by female mosquitoes during a blood meal are essential for vector reproduction and massive *Plasmodium* proliferation at the oocyst stage[7,8]. Although many studies examined associations between *Plasmodium* within-host virulence and vector fitness, only a few reported significant changes in longevity and fecundity of *Plasmodium*-infected mosquitoes[9–13]. Whether within-vector resource exploitation by parasites modulates within-host malaria virulence has never been explored.

Here, we applied theoretical and experimental approaches to examine the link between parasite within-vector resource exploitation and within-host virulence. We modeled two possible scenarios of vector–parasite symbiosis and investigated their consequences for parasite transmission dynamics. Our theoretical results predicted that vector–parasite relationship of resource acquisition shapes the dynamics of within-host virulence. While a competitive relationship maximized parasite virulence, a non-competitive relationship restricted it. To test these predictions, we experimentally blocked lipid trafficking in genetically identical mosquitoes and exploited a rodent malaria model to gauge the impact of this intervention on the within-host virulence of genetically identical parasites. We found that mosquito-derived lipids determine within-host *Plasmodium* virulence by shaping sporogony (quantity) and metabolic activity (quality) of sporozoites. By combining theoretical predictions and experimental data, we propose that the within-vector nutritional environment regulates malaria virulence. This finding has important potential implications for vector control strategies based on biological competitors or manipulation of mosquito reproduction.

## Results

### Within-vector dynamics shapes within-host malaria virulence.
To assess whether the within-vector nutrient environment could impact parasite virulence and its evolution, we used a theoretical approach that combined a within-vector dynamics model within an epidemiological framework.

We considered the within-vector resources acquired after a single blood meal as a non-replenishable nutrient source that is necessary for the development of mosquito eggs and *Plasmodium* parasites. In this sense, eggs and parasites face a direct symbiotic relationship, where the parasites either directly compete for vector nutrients (competition) or take advantage of the egg developmental dynamics to acquire vector resources (parasitism). Under competition, the parasite would benefit most if it acquired all resources, thereby preventing the mosquito from using any for its own reproduction. Under parasitism, the parasite would garner the resources initiated by the vector's oogenesis, thus benefiting from egg development.

To quantify how the quality of this symbiotic relationship (competition or parasitism) affects the development of sporozoites in the mosquito, we used the classical Lotka–Volterra model (Fig. 1a). In both symbiotic scenarios, the resulting equilibrium density of egg and parasite depends on the strength of their interactions $\alpha_1$ and $\alpha_2$ (Fig. 1a). As expected for the competitive scenario, the parasite density increased with its ability to acquire resources (i.e., with its interaction strength $\alpha_2$) at the expense of egg development, culminating in the complete castration of the vector (Supplementary Figure 1, top). In contrast, a parasitic relationship allowed for the simultaneous growth of eggs and parasites (Supplementary Figure 1, bottom), reaching a maximal parasite density when the parasite did not cause any harm to the vector (i.e., at low levels of the parasite interaction strength $\alpha_2$).

We next combined the within-vector model with an epidemiological model for vector-borne diseases describing human-to-vector and vector-to-human parasite transmission and explored how within-vector resource acquisition impacts the evolution of within-host virulence (Fig. 1b). In particular, we linked the within-vector dynamics to the epidemiological parameters describing virulence and transmission, assuming that both vector-to-human transmission and within-host virulence depend on the sporozoite density and on the efficiency of within-vector resource exploitation (i.e., the parasite interaction strength $\alpha_2$). Combining the newly defined parameters of virulence and transmission, we obtained a model of malaria epidemics as a function of within-vector processes (Fig. 1c). For a detailed description of the model see Methods.

The model allowed us to theoretically study the evolution of the parasite, exploring which vector–parasite interactions would maximize the parasite's ability to disseminate from one host to the next (i.e., parasite fitness $R_0$). To this end, we computed the parasite fitness ($R_0$; Fig. 1d, left column) and its virulence in the human host ($\nu_h$; Fig. 1d, right column) for various values of the interaction strengths $\alpha_1$ and $\alpha_2$ assuming both a competitive (Fig. 1d, upper row) and a parasitic (Fig. 1d, lower row) scenario. We found that an optimum in the parasite fitness $R_0$ was only possible in the parasitic scenario at low parasite interaction strength $\alpha_2$ (Fig. 1d, white arrows on lower row). Importantly, the observed optimal $R_0$ corresponded to low within-host virulence levels. These findings suggest that restricting within-host virulence would be beneficial for the parasite to maximize its spread in a host population. In contrast, there was no evident optimum fitness in the competitive scenario. While the parasite fitness and the within-host virulence increased with the parasite competition strength (Fig. 1d, white arrows upper row), high parasite densities led to competitive exclusion between vector and parasite, culminating at vector's extinction (Supplementary Figure 1, top).

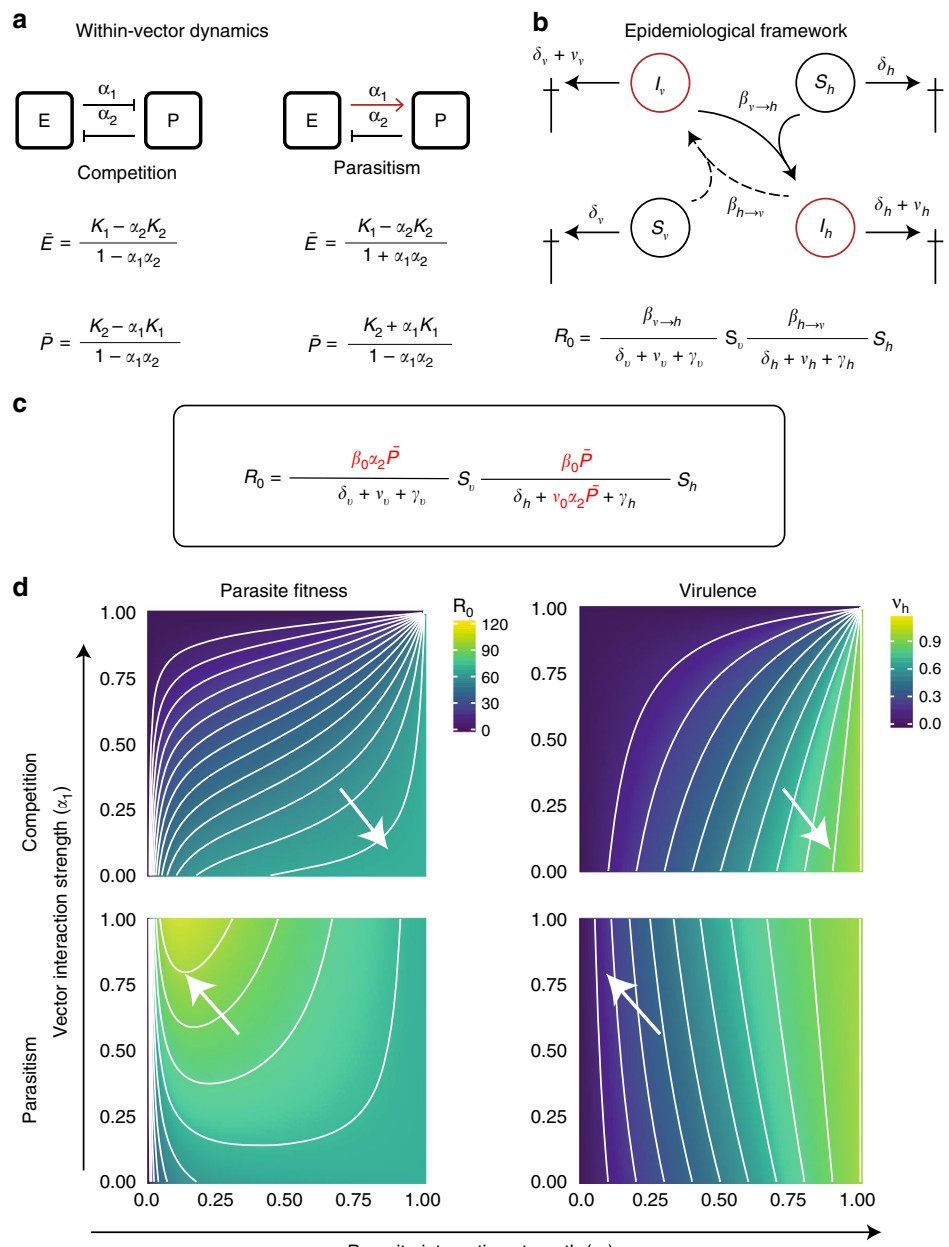

**Fig. 1** Within-vector dynamics affect the evolution of within-host virulence. **a** Schematic representation of the within-vector model. Inside the mosquito, eggs ($E$) and parasite ($P$) allocate resources for their development, interacting in an either competitive (left) or parasitic manner (right). The densities of developed parasites and eggs depend on how strongly eggs and parasites interact with each other (i.e., their interaction strengths $\alpha_1$ and $\alpha_2$) and are determined by the steady states $\bar{E}$ and $\bar{P}$. **b** Schematic representation of the epidemiological model in a human host population, where infected vectors $I_v$ (red) transmit the parasite to susceptible humans $S_h$ (black) at a rate $\beta_{v \to h}$ (solid arrows) and infected humans $I_h$ (red) transmit the disease to susceptible mosquitoes $S_v$ at a rate $\beta_{h \to v}$ (dashed arrows). For both populations, their natural death rate ($\delta_h$ and $\delta_v$) is increased by a factor ($\nu_h$ and $\nu_v$) representing the parasite-induced virulence. The fitness of the parasite is then given by the basic reproduction number ($R_0$), which is proportional to the parasite's infectivity and the duration of the infection. **c** Nested model is based on the assumption that transmission rate and virulence are affected by the interactions between eggs and parasites within the vector. **d** The fitness of the parasite ($R_0$, left column) and its virulence in the human host ($\nu_h$, right column) were calculated for various values of $\alpha_1$ and $\alpha_2$ assuming both a competitive (upper row) and a parasitic (lower row) scenario. The values of $R_0$ and $\nu_h$ are shown in the color bar from highest (lime green) to lowest (dark purple). The white arrows in the left panels point to the contours with the highest value of parasite fitness ($R_0$) and those in the right panels to their corresponding virulence values in the human host ($\nu_h$). Note that an optimal value in $R_0$ is observed only in the parasitic scenario occurring at low virulence values. The parameter values used in this model are fully described in Supplementary Table 1

In conclusion, our theoretical study predicted that the symbiotic relationship between the parasite and its vector impacts the parasite evolution. Importantly, only the parasitic within-vector relationship selected for an optimal parasite fitness at the cost of a low within-host virulence. These theoretical results prompted further questions: (1) Can we experimentally validate the principal assumption of our model that the parasite's efficiency of within-vector resource acquisition defines within-host virulence? (2) What is the nature of the within-vector symbiotic relationship? (3) What are the mechanisms underlying

the proposed symbiotic relationship?

**Mosquito lipids regulate within-host *Plasmodium* virulence.** To formalize our theoretical assumption, we evaluated whether restricting within-vector resources impacted within-host *Plasmodium* virulence and focused our studies on lipids that have been previously implicated in within-vector parasite development[14–16]. In these experiments, we interrupted lipid trafficking in adult females by RNAi-mediated depletion of the major lipid transporter lipophorin (Lp) without affecting mosquito diet or development. Lp depletion increased lipid accumulation in the midgut and inhibited ovary development (Supplementary Figure 2). We next exposed naïve mice to the bites of *Plasmodium berghei*-infected control or Lp-depleted mosquitoes. We found that Lp depletion drastically attenuated parasite infectivity and virulence. Indeed, only 20% of mice bitten by Lp-depleted mosquitoes became infected, and only 10% of them developed severe neurological symptoms of experimental cerebral malaria (ECM) compared to 90% in controls (Fig. 2a, b). However, the observed difference in the numbers of infected mice and ECM cases may result from the unequal numbers of inoculated parasites. Therefore, we subcutaneously injected naïve mice with equal numbers of sporozoites isolated from control and Lp-depleted mosquitoes. Again, fewer than 40% of mice injected with lipid-deprived sporozoites became infected, and only 20% of mice developed ECM, as compared to 100% in controls (Fig. 2c–e). Interestingly, the effect of Lp depletion was confined to the pre-erythrocytic parasite stages. Indeed, we observed similar kinetics of parasite growth after the onset of red blood cell invasion (Fig. 2f). These results confirm the principal assumption of our model and show that the within-vector lipid environment impacts within-host infectivity and virulence. Despite lipid accumulation in the midgut after a blood meal, depletion of Lp attenuated within-host infectivity and virulence of *P. berghei* parasites. These results suggest that the within-mosquito parasite development likely relies on the lipid transporter involved in oogenesis, and thus endorse the parasitic scenario.

**Mosquito lipids shape sporozoite loads and infectivity.** Our theoretical study predicted that the nature of the symbiotic relationship (competitive or parasitic) defines the evolution of within-host parasite virulence. As restricting lipid transport attenuated within-host virulence even in the absence of ovary development, we considered the parasitic scenario, where *Plasmodium* exploits mosquito Lp for lipid delivery. We microscopically gauged lipid accumulation in growing parasites using Nile Red lipid staining. In control mosquitoes, neutral lipids accumulated in the peripheral cytoplasm and vesicles of mature oocysts. In contrast, levels of Nile Red staining were significantly lower in the oocysts of Lp-depleted females (Fig. 3a, b). Moreover, the immunofluorescence analysis with anti-Lp antibodies in controls showed an increase in the proportion of Lp-positive oocysts from 25% to 72% at 7 and 13 days post infection (dpi), respectively (Supplementary Figure 3). These results support the parasitic scenario, in which *Plasmodium* exploits Lp for lipid delivery to its proliferating stages. Depletion of Lp inhibited oocyst sporulation, caused abnormal cytoplasmic vacuolizations (Fig. 3c and Supplementary Figure 4), significantly reduced the size of mature oocysts (Fig. 3d) and decreased the density of transmissible sporozoites (Fig. 3e). Similar results were obtained with the human malaria parasite *P. falciparum* (Fig. 3d, e), pointing to the conservation of mosquito lipid requirements in human and rodent *Plasmodium* species.

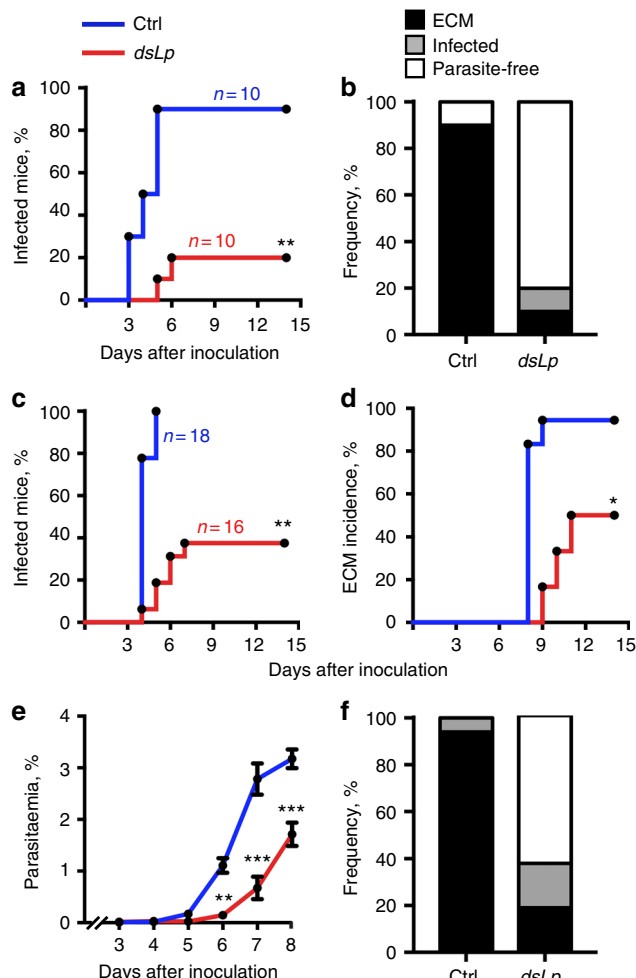

**Fig. 2** Within-vector environment impacts within-host *Plasmodium* virulence. C57BL/6J mice bitten by *P. berghei*-infected mosquitoes (**a**, **b**) or infected by sporozoite injection (**c–f**) were daily monitored for parasite occurrence in the blood and for symptoms of experimental cerebral malaria (ECM). **a** Kaplan–Meier analysis of time to infection (blood stage parasitemia) of mice infected by bites of control (Ctrl) or Lp-depleted (*dsLp*) mosquitoes (N—number of experiments: N = 4, n—total number of mice per each group: n = 10). **p < 0.001; log-rank test. **b** Cumulative health status of mice shown in **a**. **c** Kaplan–Meier analysis of time to infection (blood stage parasitemia), **d** incidence (%) of experimental cerebral malaria (ECM), and **e** parasitemia (mean ± SEM) in mice infected by subcutaneous injection of 5000 sporozoites dissected from the salivary glands of control (Ctrl) or Lp-depleted (*dsLp*) mosquitoes (N = 3, total number of mice in Ctrl and *dsLp* groups, n = 18 and n = 16, respectively). Asterisks indicate statistically significant differences (*p < 0.05; **p < 0.001; ***p < 0.0001 log-rank test (**c**, **d**) and two-sided Mann–Whitney test comparing Ctrl and *dsLp* conditions for each time point (**e**)). **f** Cumulative health status of mice shown in **c**, **d**, **e**

As within-vector lipid restriction attenuated within-host virulence after inoculation of equal sporozoite numbers, we set out to identify *Plasmodium* traits affected by Lp depletion. We first inspected sporozoite morphology and viability. Scanning electron microscopy revealed an overall normal morphology of lipid-deprived sporozoites (Supplementary Figure 5A, Supplementary Methods). We next examined whether lipid deprivation impacted sporozoite viability by comparing the proportion of live propidium iodide-negative sporozoites using image flow cytometry. In both conditions, the majority of sporozoites were PI-

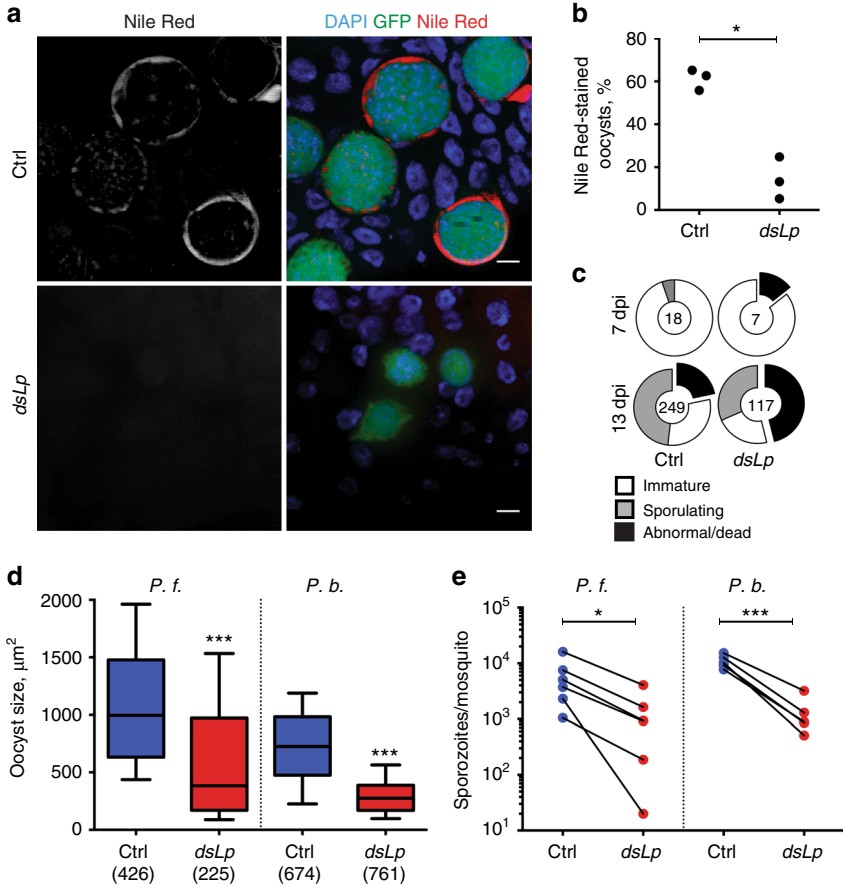

**Fig. 3** Mosquito lipids regulate within-vector parasite sporogony. Control and Lp-deficient female mosquitoes were infected with *P. berghei* (*P.b.*) (**a–c**) or *P. falciparum* (*P.f.*) (**d**, **e**) parasites. **a** Neutral lipids detected by the Nile Red stain in the GFP-expressing *P. berghei* oocysts (green) 13 day post infection (dpi) in control (Ctrl) and Lp-depleted (*dsLp*) mosquitoes. Nuclei are visualized by DAPI (blue). Scale bars—10 μm. **b** Proportion of Nile Red-positive *P. berghei* oocysts in control and Lp-depleted mosquitoes 13 dpi (N = 3, 21 mosquitoes per condition per experiment). Each dot represents mean of one experiment (*p < 0.05; paired one-sided *t*-test, paired values for *dsRNA* treatment per experiment). **c** Oocysts detected by transmission electron microscopy (Supplementary Methods) were scored according to the developmental stage: non-sporulating oocysts with normal morphology (immature, white); oocysts with normal plasma membrane retraction, budding sporoblasts, or formed sporozoites (sporulating, gray); aberrant oocysts with electron-light intracellular vacuolization (abnormal/dead, black). Proportions of the categories are shown as pie charts. The total number of analyzed oocysts per condition is indicated in the middle of each chart (N = 1 (7 dpi) and N = 3 (13 dpi), 5–14 mosquitoes per condition per experiment). The results of one representative experiment are shown. **d** Oocyst sizes of *P.f.* (11 dpi, N = 3) and *P.b.* (14 dpi, N = 6) in control and Lp-depleted mosquitoes (5–31 mosquitoes per condition per experiment). Measurements of individual mosquitoes and experiments were pooled (total numbers of oocysts are shown in brackets). The boxplot shows the median (center line), the 25th to 75th percentiles (bounds of box) and the 10th and 90th percentiles (whiskers) (***p < 0.0001; one-sided Mann–Whitney test). **e** Development of *P.f.* (14 dpi, N = 6) and *P.b.* (18 dpi, N = 6) salivary gland sporozoites in control and Lp-depleted mosquitoes (3–75 mosquitoes per condition per experiment). Each dot represents the mean number of sporozoites per mosquito per experiment. (*P.f.*) *p < 0.05, paired one-sided Wilcoxon matched-pairs signed rank test; (*P.b.*) ***p < 0.0001 paired one-sided *t*-test, paired values for different *dsRNA* treatment per experiment

negative (95%) compared to 5% in heat-killed controls, suggesting that lipid-deprived sporozoites were as viable as controls (Fig. 4a). In line with these results, the intensity of GFP fluorescence (a known marker of sporozoite viability[17] and Supplementary Figure 5B) was similar in the two conditions (Fig. 4b).

We next investigated whether lipid-deprived sporozoites showed developmental or maturation defects by comparing expression levels of key maturation markers. Again, no differences were observed between control and lipid-deprived sporozoites (Supplementary Figure 5C). Similarly, lipid deprivation had no impact on expression and surface localization of the circumsporozoite protein (CSP) involved in multiple steps of sporozoite function[18] (Supplementary Figure 5C–G). As lipids are crucial for metabolic activity, we used fluorescence microscopy and TMRE dye[19] to measure mitochondrial membrane potential. We first confirmed dye specificity by examining TMRE fluorescence after

sporozoite treatment with the inhibitor of oxidative phosphorylation CCCP (Supplementary Figure 5H). Strikingly, lipid-deprived sporozoites showed a significant decrease in TMRE fluorescence compared to controls. As this decrease was above the levels of CCCP-treated sporozoites (Fig. 4c and Supplementary Figure 5I), we concluded that Lp depletion attenuated metabolic activity of *Plasmodium* transmissible forms without affecting their viability. A broad distribution of TMRE intensities in control sporozoites compared to a sharp peak of the cytoplasmic GFP signal (Fig. 4b, c), pointed to high variability in mitochondrial activity of genetically identical sporozoites, which may contribute to the plasticity of within-host virulence. These results support our theoretical prediction that *Plasmodium* exploits mosquito resources in a non-competitive parasitic manner. Further, we show that within-vector lipid environment shapes mitochondrial membrane potential and the quantity and the quality of

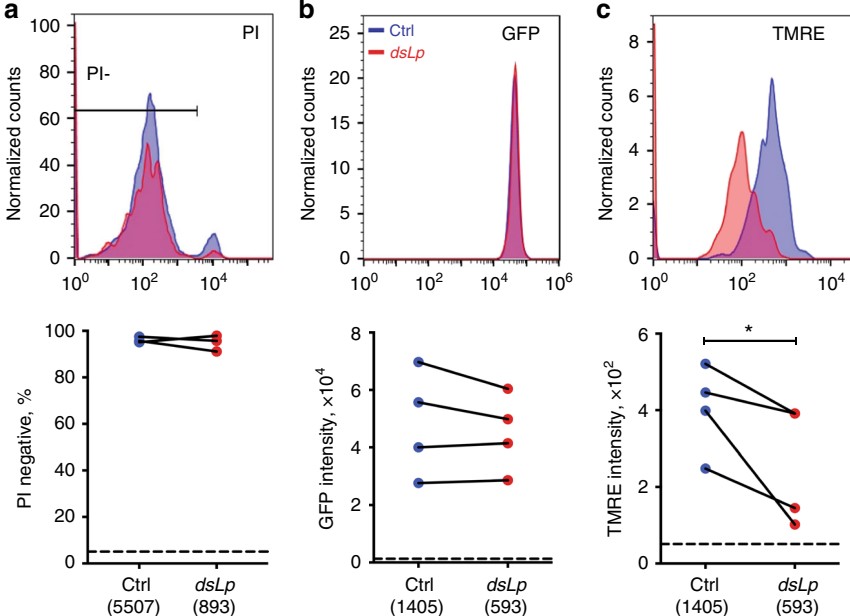

**Fig. 4** Mosquito lipids regulate sporozoite metabolic activity. **a** Imaging flow cytometry of propidium iodide (PI) staining of sporozoites. Fluorescence intensity histogram from one representative experiment (top panel). Proportion of live PI-negative sporozoites (bottom panel) for control (blue) and lipid-deprived (red) sporozoites are shown ($N = 3$). The dotted line shows the mean proportion of PI-negative heat-killed sporozoites. The total number of sporozoites are shown in brackets (12–35 mosquitoes per condition per experiment). $p = 0.317$; paired one-sided $t$-test, paired values for different $dsRNA$ treatment per experiment. Imaging flow cytometry of **b** GFP (viability) and **c** TMRE (mitochondrial membrane potential) fluorescence of control (blue) and lipid-deprived sporozoites (red). Fluorescence intensity histograms (top panels) from one representative experiment. Dot plots (bottom panels) represent geometric MFI per experiment ($N = 4$). The dashed lines show the mean MFI of heat-killed sporozoites (**b**) and of sporozoites treated with an uncoupling drug CCCP (**c**). The total numbers of sporozoites are shown in brackets (28–88 mosquitoes per condition per experiment). **b** $p = 0.1561$; **c** *$p < 0.05$; one-sided paired $t$-test, paired values for different $dsRNA$ treatment per experiment

*Plasmodium* transmissible stages.

**Time-shift between oogenesis and parasite lipid acquisition.** We next revisited the timing of lipid consumption by the mosquito ovaries and parasites and observed a clear temporal shift between these two processes. While mosquito ovaries accumulated lipids within the first 2 days after feeding[20], *Plasmodium* proliferation began 1 week post-infection (Fig. 5a). Lp depletion caused lipid accumulation in the midguts of blood-fed mosquitoes (Supplementary Figure 2). However, this lipid-rich environment after the blood meal did not benefit sporozoite fitness and attenuated the within-host infectivity and virulence even in the absence of direct competition with egg development. As only sporogonic oocysts accumulated neutral lipids (Fig. 3a), we hypothesized that early oocysts were incompetent in lipid uptake, instead, providing lipids at the initiation of oocyst sporogony should rescue parasite development and virulence.

We first tested this hypothesis by a supplemental blood feeding of mosquitoes 7 days post infection and directly gauging infectivity of the isolated sporozoites and growth of extra-erythrocytic forms (EEFs) in the hepatoma HepG2 cells in vitro. Cells were exposed to equal numbers of sporozoites isolated from control and Lp-depleted mosquitoes that received (Fig. 5f–i) or not (Fig. 5b–e) a supplemental blood feeding, at the time when the first effects of lipid starvation on the oocyst's size were observed (Fig. 3c and Supplementary Figure 4). As expected for a single blood feeding, lipid restriction attenuated sporozoite numbers, infectivity and development in vitro, as lipid-deprived sporozoites produced significantly smaller EEFs compared to controls (Fig. 5b–e and Supplementary Figure 6A). Strikingly, the supplemental feeding partially restored sporozoite numbers in

Lp-depleted mosquitoes, and completely rescued parasite infectivity and EEF growth evidenced by fluorescence microscopy (Fig. 5f–i and Supplementary Figure 6A). The supplemental feeding also restored parasite prepatency delay and ECM occurrence in mice infected with equal numbers of control and lipid-deprived sporozoites (Supplementary Figure 6B–D). Taken together, these results are in line with our hypothesis that oocysts acquire mosquito lipids after completion of vector's oogenesis.

## Discussion

Virulence is one of the most evolutionarily relevant and complex pathogen traits. Understanding virulence mechanisms provides an opportunity to predict and, potentially, control the pathogen burden and its evolution. Here, we generated a theoretical model of the mosquito–parasite relationship, experimentally evaluated it, and found that *Plasmodium* exploits the main mosquito lipid transporter for lipid delivery to the actively proliferating oocysts after completion of the vector reproductive cycle. Such time-shift in oocyst maturation may benefit the parasitic relationship by allowing a timely allocation of resources needed for host reproduction and for parasite proliferation. Temporal regulation plays a critical role in host–parasite interactions and in host immune responses. In mosquitoes, the ookinetes at the basal side of the midgut are the major target of the mosquito immune system within the first 24–48 h after blood meal, whereas the oocyst stages are spared from the complement attack[21]. Interestingly, sporogony is the longest replicative process in the life cycle of all *Plasmodium* species (12–16 days depending on the species). Given the average mosquito lifespan of 30 days, it roughly corresponds to the half-life of the mosquito Prolongation of the within-vector sojourn is risky for the parasite, which should be

selected to maximize its chances to infect the next host as fast as possible. Therefore, the beneficial effects of lengthy sporogony must be under a strong selective pressure. Currently, the processes that regulate parasite's competence to acquire lipids remain unknown. Perhaps, the extracellular location of the oocysts imposes some restrictions on the rate of parasite sporogony as opposed to intracellular sporogony of red blood stages. Future comparative studies of transcriptional profiles of young and sporogonic oocysts may reveal new players in this process, whose

manipulation will be essential to confirm or refute our hypothesis. Regardless of the mechanisms in place, we predict that shifting the timing of resource allocation should have dramatic consequences for parasite development, transmission, and virulence.

Our observations suggest that limited lipid resources during sporulation induce an autophagy-like cell death. Autophagy, often characterized by cytoplasmic vacuolization, has been observed in a large proportion of *Plasmodium* liver stages[22], and our results extend these observations to the extracellular oocysts.

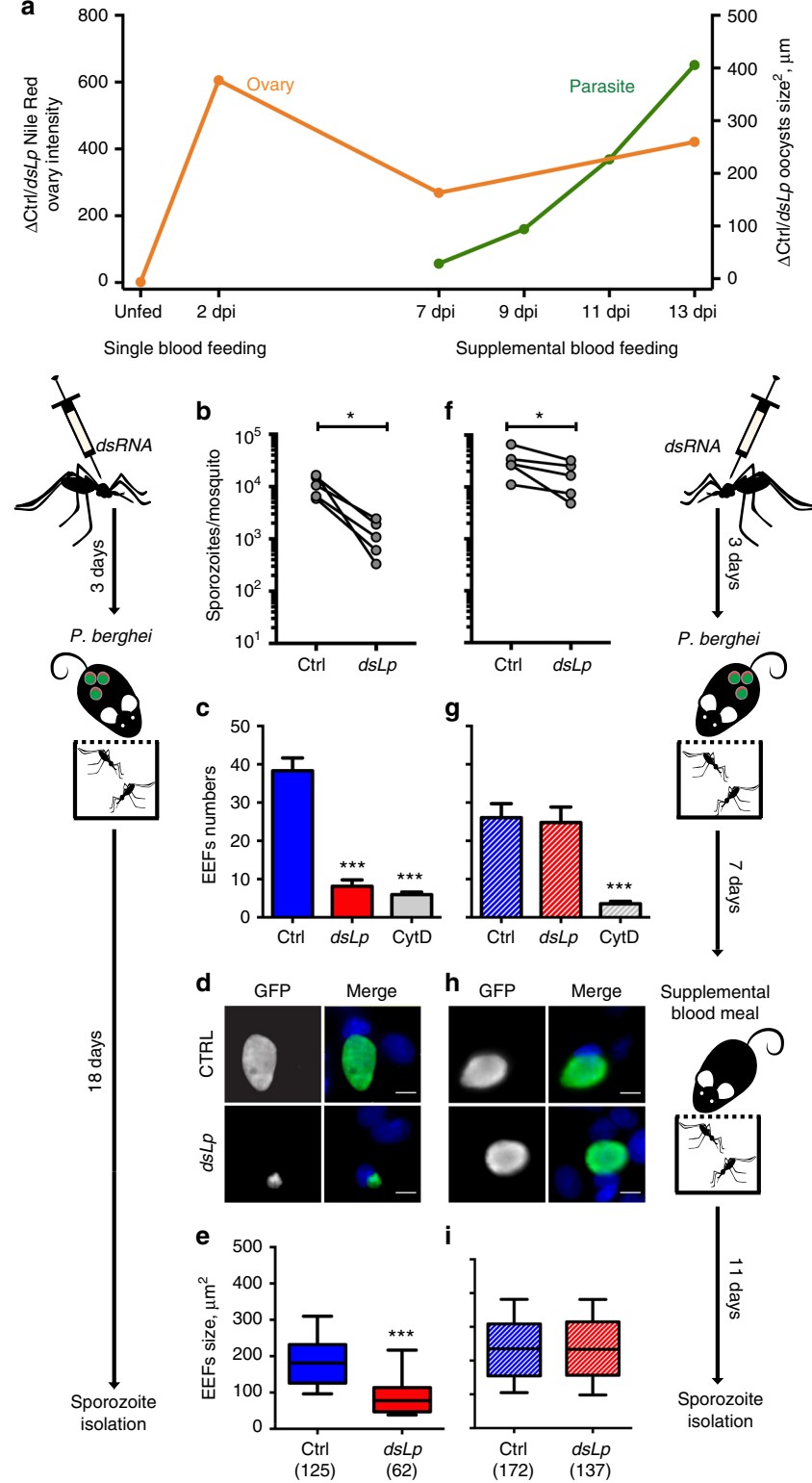

Excessive numbers of oocysts per midgut could cause nutrient restriction and curb parasite development[23]. In contrast, in our experiments, inefficient lipid trafficking inhibited the development of similar numbers of the pre-sporogonic oocysts[15]. Interestingly, extended nutrient deprivation triggers cell autophagy and induces mitochondrial dysfunction in mouse embryonic fibroblasts[24]. Similarly, we found that lipid restriction blunted sporozoite's mitochondrial membrane potential. Mitochondria are essential for mosquito stages of the *Plasmodium* life cycle, as mutations in mitochondrial genes arrest growth and sporulation of oocysts[25–29]. Consistently, inhibition of mitochondrial potential by the antimalarial drug atovaquone inhibits *Plasmodium* sporogonic development and reduces the number and size of developing EEFs in vitro[30–32]. Together with these findings, our results point to mitochondria as crucial regulators of within-vector *Plasmodium* proliferation, transmission, and within-host virulence. Although drug pressure on the mitochondria frequently selects for resistance, most of the identified mutations are loss-of-function, that disrupt within-vector parasite development and disable transmission[28]. Therefore, *Plasmodium* mitochondria may represent an evolution-proof drug target[33], as long as no gain-of-function mutations emerge that may increase mitochondrial activity and, thereby, parasite virulence. Altogether, our data demonstrate that within-vector environment determines within-host virulence by shaping sporogony (quantity) and metabolic activity (quality) of sporozoites.

At the ecological level, our results suggest that the plasticity of the within-host parasite virulence is metabolically regulated. We found that even in laboratory conditions, sporozoites display broad variability in their mitochondrial activity. This implies that mosquito life-history traits can directly impact the within-host parasite virulence. Variable trophic environments of larval breeding sites[34] and blood meals should translate individual mosquito nutritional variance into variability in the within-host *Plasmodium* virulence. Indeed, multiple reports have implicated larval diet and adult blood meals in determining parasite loads within a mosquito[35–40], but this has not been linked to within-vector modulation of parasite metabolic activity and within-host virulence. Our findings open up a new perspective on mosquito transmission capacity and on evolution of malaria virulence.

The evolutionary responses to vector control measures are often non-linear due to the complexity of vector–parasite relationships. Our theoretical approach predicts that competitive within-vector *Plasmodium* dynamics may exacerbate within-host virulence, potentially leading to vector sterilization and elimination. The model indicates that parasitic exploitation of within-vector resources selects for an optimal parasite fitness ($R_0$) by restricting the evolution of *Plasmodium* within-host virulence. Importantly, harm towards the vector's reproduction alone does not restrict virulence evolution[41]. Based on these predictions, we provide empirical evidence that within-vector *Plasmodium* development requires neutral lipids carried by the major vector lipid transporter. Our hypothesis of a time-shift between vector oogenesis and parasite sporogony is consistent with the parasitic scenario. Indeed, the parasites may develop competence for lipid uptake after the completion of egg development 1 week after infection. This profitable developmental niche is retained as long as nutrient supply is sufficient. Changes in the nutrient environment should similarly impact oogenesis and parasite sporogony, explaining the positive correlation between ovary development and parasite loads often observed in experimental infections. Further investigation of mosquito and parasite factors involved nutrient acquisition and metabolism is necessary to predict potential switches between parasitic and competitive vector–parasite relationships. Such situations could arise from targeting mosquito reproduction factors by transgenic approaches[42], or from the introduction of biological competitors, such as *Wolbachia*[43]. Altogether, our data highlight the links between metabolism and life-history traits of vector and parasite, and their potential in shaping evolution of malaria virulence. Further theoretical, experimental, and field studies should address the effects of natural and artificial perturbations of vector and parasite metabolism on malaria epidemiology.

## Methods

**Model of within-vector and epidemiological dynamics**. The resources acquired by a single blood meal (non-replenished nutrient source) are used by both vector and parasite for, respectively, reproduction and sporozoite development. To explore the quality of resource dependence for pathogen and egg development within the vector, we employ the competitive Lotka–Volterra equations[14,15], modeling the inter- and intra-specific competition between mosquito eggs ($E$) and parasites ($P$):

$$\dot{E} = r_1 E \left( 1 - \frac{E + \alpha_2 P}{K_1} \right) \qquad (1)$$

$$\dot{P} = r_2 P \left( 1 - \frac{P + \alpha_1 E}{K_2} \right). \qquad (2)$$

Here, $r_1$ and $r_2$ represent the maximum intrinsic growth rate of eggs and parasites, respectively, and $\alpha_1$ and $\alpha_2$ describe the strength of their reciprocal interaction. The maximum egg and sporozoite densities are given by the carrying capacities $K_1$ and $K_2$.

To consider parasitic interactions, we modify Eq. (2) as follows:

$$\dot{P} = r_2 P \left( 1 - \frac{P - \alpha_1 E}{K_2} \right). \qquad (3)$$

---

**Fig. 5** A time-shift between vector oogenesis and parasite lipid acquisition. **a** Lipid acquisition by mosquito ovaries precedes *P. berghei* oocyst development. The ratios of Nile Red mean intensities in the ovaries (orange line, $N = 3$) and oocysts growth (green line, $N = 2$) between control and Lp-depleted mosquitoes are shown. Mean oocyst sizes and Nile Red ovary intensities from pooled mosquitoes and experiments ($N \geq 2$, 5–13 mosquitoes per time point per condition per experiment). **b** Lp-depleted mosquitoes (*dsLp*) and controls (Ctrl) were infected with *P. berghei* and offered (**f–i**) or not (**b–e**) a supplemental blood meal 7 days post infection (dpi) (3–63 mosquitoes per condition per experiment). **b**, **f** Salivary gland sporozoites were isolated and counted from control and Lp-deficient mosquitoes. Each dot represents the mean number of sporozoites per mosquito per experiment ($N = 5$; *$p < 0.05$; one-sided paired *t*-test, paired values for different *dsRNA* treatment within the same experiment). **c**, **g** Human hepatoma HepG2 cells were infected with 10,000 sporozoites and the number of extra-erythrocytic forms (EEFs) per field was gauged by microscopy 2 dpi. As a negative control, sporozoites from control mosquitoes were treated with cytochalasin D (Cyt D). Five microscopic fields were imaged per well per condition in triplicates, except for two cases for *dsLp* due to the low number of sporozoites. Mean number of EEFs of pooled experiments ($N = 3$) and SEM are shown (***$p < 0.0001$; Kruskal–Wallis test and Dunn's multiple comparison test). **d**, **h** Representative fluorescent micrographs of GFP-expressing EEFs (green) developing in HepG2 cells. Cell nuclei are stained with DAPI (blue). Scale bar—10 μm. **e**, **i** Growth of EEFs produced by sporozoites from control and Lp-deficient mosquitoes. EEFs sizes of pooled experiments are shown ($N = 3$). The boxplots show the median (center line), the 25th to 75th percentiles (bounds of box) and the 10th and 90th percentiles (whiskers). The total numbers of EEFs are shown in brackets. **e** ***$p < 0.0001$; **i** $p = 0.4223$; one-sided Mann–Whitney test

Depending on the parameter values, these classical models can describe three qualitatively different scenarios: extinction of both species, competitive exclusion, or co-existence. We here focus on the non-trivial steady states (i.e., co-existence), which only occur if both growth rates $r_1$ and $r_2$ are positive and neither species outcompetes the other, i.e., if $0 < \alpha_1 < 1$ and $0 < \alpha_2 < 1$. The non-trivial steady states for both models, i.e., competition and parasitism, are given in Fig. 1a. The parameters are fully described in Supplementary Table 1.

The simplest model of malaria epidemics considers the populations of infected humans $I_h$ and vectors $I_v$ as given by the following set of differential equations:

$$\dot{I}_h = \beta_{v \to h} S_h I_v - (\delta_h + \nu_h + \gamma_h) I_h \tag{4}$$

$$\dot{I} = \beta_{h \to v} S_v I_h - (\delta_v + \nu_v + \gamma_v) I_v. \tag{5}$$

In this model, infected vectors $I_v$ transmit the parasite to susceptible humans $S_h$ at a rate $\beta_{v \to h}$. Humans recover from the disease at a rate $\gamma_h$, and their natural death rate $\delta_h$ is increased by a factor $\nu_h$ representing the parasite's detrimental effect on its host.

Similarly, infected humans $I_h$ transmit the disease to susceptible mosquitoes $S_v$ at a rate $\beta_{h \to v}$. Vectors recover at a rate $\gamma_v$, and their natural mortality $\delta_v$ increases by the parasite-induced virulence $\nu_v$. Importantly, this model assumes that the population of both susceptible humans and vectors remain constant.

In epidemiology, the basic reproduction number $R_0$ represents a key parameter used to study the fitness of an infectious agent, i.e., whether it can invade and persist in a population. Classically defined as the number of secondary infections produced by an infected individual in an otherwise susceptible host population, $R_0$ is proportional to the parasite's infectivity (i.e., transmission rate) and the duration of a typical infection[44]. As we here consider an infection that involves two hosts, the total $R_0$ can be described as follows:

$$R_0 = \frac{\beta_{h \to v}}{\delta_v + \nu_v + \gamma_v} S_v \frac{\beta_{v \to h}}{\delta_h + \nu_h + \gamma_h} S_h. \tag{6}$$

Importantly, a parasite can maintain itself in a population only if $R_0 > 1$.

To study how the resource allocation for parasite and eggs development within the mosquito affect the transmission of the parasite in a human population, we next link the within-vector dynamics to the epidemiological parameters describing virulence and transmission.

Because vector-to-human transmission rate increases with the number of produced sporozoites[45], we first postulated that vector-to-human transmission rate is a linear function of equilibrium parasite density, i.e., $\beta_{v \to h}(P)$. Furthermore, within-vector resource competition may also affect the parasite's capacity to infect the human host[23,39,46]. Therefore, we assumed that $\beta_{v \to h}(P)$ increases with the efficiency of resource exploitation, i.e., with parasite's competition coefficient $\alpha_2$. Assuming that the sporozoite production within the mosquito reaches its equilibrium before the time of transmission, the vector-to-human transmission rate is described as $\beta_{v \to h}(\bar{P}) = \beta_0 \alpha_2 \bar{P}$, where $\beta_0$ is the parasite's basic transmission rate, and $\bar{P}$ is the parasite equilibrium density as given in Fig. 1a.

Host-to-vector transmission depends on the parasite's asexual replication and production of sexual forms (gametocytes)[5,47]. We here consider that the capacity to enter the asexual cycle depends on initial parasite load, resulting in a human–vector transmission $\beta_{v \to h}(\bar{P}) = \beta_0 \bar{P}$.

Finally, we assume that the detrimental effect human experiences during infection (i.e., virulence) is proportional to the overall replication rate of asexual parasites. Further considering that the efficiency of resource exploitation within the mosquito also affects the parasite's capacity to replicate in the human host, the total parasite-induced virulence in humans is $\nu_h(\bar{P}) = \nu_0 \alpha_2 \bar{P}$. Of note, assuming that virulence saturates depending on the initial parasite load (i.e., $\nu_h(\bar{P}) = \nu_0 \alpha_2 \bar{P}/(h + \bar{P})$, where $h$ represents the parasite density at which the virulence is half maximal) does not qualitatively affect our results, as shown in Supplementary Figure 7.

Combining the newly defined parameters for virulence and transmission into Eq. (6), we obtain the final model of malaria epidemics as a function of within-vector processes:

$$R_0 = \frac{\beta_0 \bar{P}}{\delta_v + \nu_v + \gamma_v} S_v \frac{\beta_0 \alpha_2 \bar{P}}{\delta_h + \nu_0 \alpha_2 \bar{P} + \gamma_h} S_h. \tag{7}$$

This model allows us to study the evolution of the parasite, i.e., study under which vector–parasite interactions (competitive or parasitic), $R_0$ will be maximized. The parameters of the full model are provided in Supplementary Table 1.

**Mosquito rearing and parasite infections.** *Anopheles gambiae s.l.* mosquitoes were used throughout the study. To prevent large losses at early parasite stages (ookinetes) inflicted by the mosquito complement-like system in the absence of Lp[15,48], all infection experiments were performed with immunodeficient mosquitoes (*7b* line)[49]. *7b* mosquitoes, were maintained at 29 °C 70–80% humidity 12/12 h

day/night cycle. In *P. falciparum* infections, mosquitoes were fed for 15 min on a membrane feeder with NF54 gametocytes cultured with O+ human red blood cells (Haema, Berlin), and, thereafter, kept in a secured S3 laboratory according to the national regulations (Landesamt für Gesundheit und Soziales, project number 411/08). The NF54 clone used in this study originated from Prof. Sauerwein's laboratory and was authenticated for *Pfs47* genotype by PCR on genomic DNA. *P. falciparum* asexual cultures were monthly tested for *Mycoplasma* contamination. For *P. berghei* infections, mosquitoes were fed on anesthetized CD1 mice infected with the GFP-expressing *P. berghei* GFP-con 259cl2 clone[50] (ANKA strain, provided by Dr. Blandine Franke-Fayard). Shortly after infections, unfed mosquitoes were removed, fed mosquitoes were maintained at 26 °C for 11–14 days (*P. falciparum*) or at 20 °C for 7–19 days (*P. berghei*), and then used for the midgut and/or the salivary gland dissections.

**RNAi silencing.** A 424 bp long fragment of *lipophorin* gene (AGAP001826) and a 818 bp long fragment of the *LacZ* gene (sequences in Supplementary Table 3) were cloned into the pLL10 vector to generate pLL345 and pLL100, respectively[15]. Double stranded RNA (dsRNA) was produced using the MEGAscript T7 Transcription kit and purified using MEGAclear Transcription Clean-Up Kit (Ambion). For RNAi silencing, 1–2 day-old females (200–400 per condition per experiment) were anesthetized with $CO_2$ and injected with 69 nl of 3 µg µl$^{-1}$ *dsLacZ* (control) or *dsLp* using a Nanoject II Injector (Drummond). Mosquitoes recovered for 3–4 days following injection before infections. Efficiency of RNAi silencing is summarized in Supplementary Figure 8.

**Immunofluorescence analysis.** Mosquito midguts were dissected in PBS, fixed in 4% formaldehyde, and washed in PBS. Tissues were permeabilized with 0.1% Triton X-100 and incubated with a 1:1 mix of mouse monoclonal anti-ApoI and anti-ApoII antibodies (1:300, clones 2H5 and 2C6)[15] overnight at 4 °C followed by incubation with the secondary AlexaFluor555-labeled antibodies at 1:1000 (Molecular Probes, A-21422), or with 0.1 µg ml$^{-1}$ of the Nile Red stain for 40 min at room temperature (Sigma Aldrich). Nuclei were stained with DAPI (1.25 µg ml$^{-1}$, Molecular Probes) for 40–60 min at room temperature. Images were acquired using an AxioObserver Z1 fluorescence microscope equipped with an Apotome module (Zeiss). Oocyst size was measured by hand-designing a circular ROI of randomly selected oocysts (identified by GFP for *P. berghei* or by bright field and DAPI staining for *P. falciparum*) using ZEN 2012 software (Zeiss). Images were then processed by FIJI software (ImageJ 1.47 m).

**Development of *P. berghei* liver-stages in vitro.** Sporozoites from the dissected salivary glands of *P. berghei*-infected mosquitoes (18–19 dpi) were collected into RPMI medium (Gibco) containing 3% bovine serum albumin (Sigma Aldrich) and enumerated with a hemocytometer. The *P. berghei* liver stages were cultured in vitro in HepG2 hepatoma cells and analyzed using standard techniques[51]. HepG2 cells were originally provided by Prof. K. Matuschewski and were tested for *Mycoplasma* contamination. Briefly, 15,000–20,000 HepG2 cells per well (70% confluence) were plated on transparent-bottom 96-well plates (Nalgene International) and incubated for 24 h, then seeded with 10,000 sporozoites and co-cultured at 37 °C for 2 h. Free sporozoites were removed by washing. As a negative control, sporozoites isolated from control mosquitoes were pre-treated with the inhibitor of actin polymerization cytochalasin D (Sigma Aldrich) for 10 min. At 48 h post seeding, cells were fixed with 4% formaldehyde for 10 min, washed with PBS and blocked with 10% fetal calf serum in PBS. Development of the liver-stage parasites was examined using monoclonal rabbit anti-GFP antibodies (1:1000, AbCam, ab290) and secondary AlexaFluor488 conjugated anti-rabbit antibodies (1:1000, Molecular Probes, A-11008). Nuclei were stained with DAPI (1.25 µg ml$^{-1}$, Molecular Probes). Images were recorded directly on the 96-well plate using an AxioObserver Z1 fluorescence microscope equipped with an Apotome module (Zeiss) and analyzed for number and size of EEFs using the Axio-Vision ZEN 2012 software (Zeiss).

**P. berghei mouse infections in vivo.** Mice were housed and handled in accordance with the German Animal Protection Law (§8 Tierschutzgesetz) and both institutional (Max Planck Society) and national regulations (Landesamt für Gesundheit und Soziales, registration number H 0027/12). C57BL/6J mice were purchased from Charles River until 2016 (Fig. 2) or bred in the institute animal core facility from 2017 (Supplementary Figure 6). To determine infectivity to mice, the sporozoites collected from the mosquito salivary glands (18–19 dpi) were injected subcutaneously (5000 sporozoites per mouse) into the tails of 8–10-week-old C57BL/6J females. Bite-back experiments were performed by challenging anesthetized naïve C57BL/6J mice with *P. berghei*-infected mosquitoes (18 dpi, 13–18 mosquitoes per mouse per group). Parasitemia was determined by daily Giemsa staining of thin blood smears and FACS analysis of the red blood cells with the GFP-expressing blood stage parasites. Infected mice were monitored every 6–12 h for the appearance of severe neurological and behavioral symptoms typical of ECM such as hunched body position, grooming

alteration, ataxia, paralysis, or convulsions[52] or by the rapid neurological and behavioral test (RMCBS)[53]. All mice with ECM symptoms or the RMCBS score equal or below five out of 20 were sacrificed immediately.

**Sporozoite imaging flow cytometry**. Salivary gland sporozoites (18–19 dpi) were isolated into RPMI medium (Gibco) 3% bovine serum albumin (Sigma Aldrich) and kept on ice until staining. Sporozoites were diluted to 3 mosquito equivalents in 150 μl 1× PBS and stained as described below. For viability measurements, sporozoites were stained with propidium iodide (10 μg ml$^{-1}$, Sigma Aldrich) for 5 min at room temperature[54]. For CSP staining, sporozoites were incubated for 30 min at 4 °C with AlexaFluor647-conjugated anti-PbCSP mouse monoclonal antibodies (clone 3D11, BEI Resources, 1.2 μg ml$^{-1}$). Total sporozoites were acquired after one wash in PBS (16,000×$g$, 4 min, 4 °C) using an ImageStreamX Mk II (Merck Millipore) with a 60× objective for 40 min. For membrane potential measurements, sporozoites were stained with TMRE (5 nM, Cell Signaling Technology) for 20 min at 20 °C. GFP-positive sporozoites were acquired without washing using an ImageStreamX Mk II (Merck Millipore) with a 60× objective for 1 h. The analysis was performed with IDEAS 6.2 (Merck Millipore) and FCS files were exported and analyzed by FlowJo v10. Given the small size of the mitochondria, the feature finder tool from IDEAS software was used to improve gating of optimally focused sporozoites. The following parameters were used to discriminate in-focus single sporozoites from debris, and to identify in-focus sporozoites to resolve their mitochondria: H Entropy Std_M01_Ch01_15 over H Variance Std_M01_Ch01, Aspect Ratio_M02 over Symmetry 4-Object(M02_Ch2) Ch02, Gradient RMS-M01-Ch01 over H Correlation Mean-M02-Ch02_3, and Gradient RMS-M02_Ch02. Briefly, these parameters measure brightfield image texture, GFP image shape, brightfield and GFP focus, respectively. For quantification of mitochondrial membrane potential and GFP intensity, each sample was time-gated to exclude artefacts of prolonged staining. Moreover, to avoid a possible bias due to the variable pre-acquisition waiting times on ice, the order of sample acquisition was swapped between the experimental replicates. The parameter corresponding to the lowest background signal in unstained sporozoites was Bright Detail Intensity R7_M04_Ch04, which was selected as bona fide TMRE intensity readout. For GFP, the Intensity_MC_Ch02 was used, while for propidium iodide the Intensity_MC_Ch04 was used. Heat-killed (65 °C for 5 min) sporozoites and sporozoites treated with a mitochondrial uncoupling drug CCCP (Cell Signaling Technology, 50 μM for 20 min at RT) were used to validate PI and TMRE as markers for sporozoite viability and membrane potential, respectively.

**Statistical analyses**. No animals or samples were excluded from the analyses. Mice and mosquitoes from the same batches were randomly allocated to the experimental groups (age range: 1–2 days for mosquitoes and 1 week for mice). The experimenters were not blinded to the group allocation during the experiment and/or when assessing the outcome. Sample sizes were chosen according to best practices in the field and previous data[15,16], and are indicated in Supplementary Table 2.

Statistical analysis was performed with GraphPad Prism 7 and R Studio Version 1.1.453 software and $p$-values below 0.05 were considered significant (*$p < 0.05$; **$p < 0.001$; ***$p < 0.0001$) as indicated in the figures. All datasets were first tested for normality (Shapiro–Wilk Normality Test) and homogeneity of variances (Levene's Test), and statistical tests were applied accordingly. The specific tests used are indicated in the figure legends.

**Code availability**. The within-vector and epidemiological framework were implemented in the R programming language, version 3.4.2. The used scripts are available under https://github.com/levashinalab/resourceExploitation.

**Data availability**. The data sets generated during and/or analyzed during the current study are available from the corresponding author on request.

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

## Acknowledgements

This work was supported by Max Planck Society, EC FP7 EVIMalaR (Grant Agreement No. 242095) and MALVECBLOK (Grant Agreement No. 223601). The authors thank H. Krüger, M. Andres, and L. Spohr for mosquito rearing, mouse work assistance, and *Plasmodium* infections, and D. Tschierske and D. Eyermann for *P. falciparum* cultures. The authors express their gratitude to Prof. K. Matuschewski and his group for continuous fruitful discussions, to Dr. M. M. Mota for helpful comments. The following reagent was obtained through BEI Resources, NIAID, NIH: Hybridoma 3D11 Anti-*Plasmodium berghei* 44-Kilodalton Sporozoite Surface Protein (Pb44), MRA-100, contributed by Victor Nussenzweig.

## Author contributions

G.C., M.G. and E.A.L. conceived the study and designed the experiments. M.G. and P.C. B. performed the modeling analysis. G.C., M.E., R.L., C.G. and V.B. performed experiments. A.E.H. and R.S. contributed reagents and expertise. G.C., M.E., M.G., R.L., R.S., C. G., V.B., P.C.B. and E.A.L. analyzed the data. G.C., M.G., P.C.B. and E.A.L. wrote the manuscript.

## Additional information

**Competing interests:** The authors declare no competing interests.

