## [Peer Review File · Nature Communications]

Reviewers' comments:

Reviewer #1 (Remarks to the Author):

The manuscript tackles the long-neglected topic of within-vector ecology in a novel way, revealing unexpected and important biology. The authors are preaching to the converted when they contrast the depth of understanding of within-host dynamics and how selection has shaped parasite phenotypes expressed in the host to analogous information for the vector phase of the life cycle. Targeting parasite phenotypes in the vector offers novel intervention opportunities, but first, an understanding of such phenotypes in the context of both within- and between-host processes is required. The authors go a long way towards this by revealing a novel phenotype that could both block transmission and reduce virulence to humans.

Specifically, the finding that the within-mosquito environment modulates infectivity and virulence to the next host is the most striking result and I am sure it will have a significant impact on the field. Tracking the phenotype down to lipid availability is impressive too. Whilst on the whole I find the data to be compelling (I don't really have the expertise to comment on the technicalities of the model), I think their presentation could be made stronger and more accessible to a general audience. Thus, my comments at this stage are mostly fairly general.

1) The terminology and framework used in the model is rather confusing and only makes sense in light of the data. I would either rename the strategies or explain the logic of the nomenclature more clearly in the introductory text. The idea seems to be that mosquitoes get resources from blood that can be used for reproduction or used by the parasite. In a "competitive" scenario, the mosquito and parasite compete directly for resources as if they are free-living organisms competing over external resources. This view might be fine mathematically but is very confusing because the resources are actually owned by the mosquito; the mosquito is responsible for processing the blood into resources that are accessible to the parasite. Thus, this scenario is a form of parasitism and biologically hard to differentiate from the "parasitic scenario". I wonder if reading Clay Cressler's work (eg <http://onlinelibrary.wiley.com/doi/10.1111/ele.12229/epdf>) would be useful to help frame these scenarios more clearly?

- similarly, the last para of the discussion is quite confusing. It would help to be clear when virulence is mentioned, whether its to the host or vector.

2) Moreover, I am not entirely convinced the data can differentiate the model's scenarios and so I find it distracts me from the main strength of the study which is the very interesting data and implications. Perhaps either removing the model or presenting it after the data in the context of the results motivating theoretical exploration of the epidemiological consequences of a potential explanation for the delay in lipid uptake, will make the paper flow better and the strategies be easier to convey?

3) It is clear that lipid depleted parasites are less infective, but think it might be premature

to conclude their phenotype as less virulent in the next host. Even though the number of sporozoites is controlled for in some of the experiments, it could still be a function of infective dose. If a smaller proportion of sporozoites from lipid depleted vectors are healthy, then infections are effectively being started with fewer viable parasites. Is there evidence that even very very low numbers of "normal" sporozoites usually cause cerebral symptoms? This would add weight to the notion that something more than simply a reduction in the number of healthy sporozoites is happening.

4) I have a few concerns about the analyses used, presumably due to the lack of detail it has been possible to include.

a- The sample sizes for mice in the experiments are not clear. I assume the n in Fig2 for A-C and D-F are number of mice monitored (it reads as if referring to numbers of mosquitoes dissected)?

b- Also, its not obvious what the number of mosquitoes per condition per experiment is. Eg Fig3 and 4 could be clearer in whether n refers to number of replicate experiments or mosquitoes within a condition. Just giving the number of oocysts is not helpful because mosquitoes or experiments are the independent replicates (see 4d).

c- Being an evolutionary biologist, we are more picky about analyses than parasitology generally is. I'm worried about the use of parametric tests on n=3 samples. There are clear to see effects in the plots so it is unlikely type 1 errors are being made, but the using an anticonservative analysis may give false confidence in effect sizes.

d- It looks like experiment means are presented in most figures but it is not clear whether data for individual oocysts, mosquito means, or experiment means are used in analyses. Whilst assaying lots of oocysts in each mosquito is better than assaying a few, it is not the case that each oocyst is an independent data point. Strictly speaking, each mosquito is a sample and the more oocysts examined, the more accurate the means are for each mosquito. In other words, oocysts sampled from the same mosquito are not independent of each other. The best way to analyse such data is to either "summarise" the data for each mosquito as its mean, and so groups of means can be compared with e.g. T tests or ANOVA. Or, with mixed effects models that use the data for each oocyst whilst accounting for the fact they belong to specific mosquitoes and different experimental blocks. If the authors have not used either of these approaches there could be issues of false confidence in effect sizes.

e- Am I right in thinking the interaction plots and paired T-tests pair treatments within each experimental block? If so, clarifying this in the figure legends would be useful.

5) I'm not convinced by conclusion that the ~1 week delay in lipid use by the parasite is explained as the parasite evolving to be restrained. I think the scenarios below merit consideration and discussion. As the authors state, determining the correct scenario matters for the evolutionary consequences of interventions that target lipid availability.

a- The idea that the mosquito is allowed to reproduce by the parasite and then allows its remaining lipids to be used by the parasite is not the conclusion I would come to first. It's not obvious to me that the parasite is restraining because the host could be in control and (somehow) not allowing lipids to be accessible to parasites for this time. Also, I'd assume the mosquito would have a use for these left-over lipids, for example, for subsequent bouts of reproduction. So, there could still be a "competitive relationship" but the vector is "winning".

b- If it really is parasite restraint, then parasites should still show the delay in lipid extraction even in mosquitoes that have an excess of lipids (is it possible to give a blood meal with enhanced lipids, or infecting mosquitoes after their first egg lay when they have spare lipids?).

c- It could be a cost of crowding (this would still be very interesting): the availability of lipids determines the number of sporozoites that can successfully be matured. If so, it is not a parasite strategy but an unavoidable consequence of an environmental constraint. The authors say "Nutrient restriction in natural conditions could be induced by oocyst crowding; however in our experiments the numbers of pre-sporogonic oocysts between the experimental groups were similar." I don't see how these comments relate. The same numbers of oocysts mean that those in depleted mosquitoes are resource limited; ie fewer resources per capita.

d- Furthermore, claiming a phenotype to be an adaptive strategy generally requires more support. Eg showing that parasites ultimately lose out on fitness if they use resources sooner. As an evolutionary biologist, one immediately considers a competitive scenario: if multiple parasite genotypes are present in the mosquito, the genotype that extracts lipids first does best (because it gets more lipids than its competitor) which results in selection for an arms race for earlier lipid extraction. Given that mosquitoes without lipid reserves don't seem to have short lifespans, there won't be a trade-off with premature death of the vector to constrain the timing of lipid extraction. Also, the parasites interests would be best served by scavenging lipids for the purpose of stopping the mosquito from reproducing. It has become clear that infection reduces mosquito lifespan but only when mosquitoes reproduce. Thus, if the parasite were able to extract lipids earlier they might mature a greater proportion of healthy sporozoites and extend the lifespan of the vector.

e- Having said the above, there is an evolutionary scenario (Mafia hypothesis, Restif & colleagues) that could apply. In this scenario, the host gives the parasite resources in return for restraining its virulence.

6) Are there data on egg output? It would be nice to show that infected control mosquitoes do actually reproduce. If they don't, then the hypothesis that the parasite is restraining to allow reproduction of the vector is not supported.

7) I don't believe the data shows the evolution part of the statement that "that evolution of malaria virulence is metabolically regulated by the within-vector nutritional environment". Within-vector resources are a mechanism that regulates virulence. Demonstrating the

“evolution” part would require revealing eg signatures of selection on genes involved in resource extraction or a change in these processes if within-host virulence is experimentally perturbed in an artificial selection regime. This is just an example of precision in evolutionary terminology and does not detract from the value of the results.

8) It would increase the impact for a general audience if the results for malaria infection could be set in the context of other diseases showing similar resource use strategies? Eg is “parasite restraint” a common strategy in nature or are Plasmodium seemingly novel in this regard? Including such info in the intro would provide valuable context for why the impact of within-vector processes on within-host processes is investigated.

9) A final thought is Spence et al Nature showed mosquito transmission per se attenuates virulence, probably via eliciting different immune responses to blood passaged parasites. Is it possible that in normal circumstances*, some lipid deprived parasites will always occur in mosquitoes and could be responsible for eliciting the phenotype Spence et al observed? (*eg species that do not replicate to such high numbers in mosquitoes)

Sarah Reece

Reviewer #2 (Remarks to the Author):

The manuscript presents both theoretical and experimental results exploring how within-vector resource competition impacts malaria virulence. The theoretical work shows that if there is straight competition between the vector and parasite for resources that there will be runaway virulence, but that if the relationship is truly parasitic then virulence can be stabilised. They then show experimentally that the parasite may be avoiding competition by delaying its replication.

I should say at the outset that I am a theoretician, and that I therefore do not feel fully confident assessing the experimental procedures. I thought the manuscript examined an important question, the methodologies (both theory and experiment) appear well planned, they find a novel result and the manuscript itself is well-written. The combination of theory and experiment is a real strength, and identifying how malaria parasites may be avoiding resource competition is interesting both theoretically but also from a treatment perspective. I have focussed my comments below on the theoretical model.

My only real question is to do with the fact that so many of the modelled processes are linear. That vector-to-human transmission (eq 5) is linear in P is fairly common (though vector-borne diseases are often modelled as being frequency-dependent), but the justification for virulence (eq 6) and competition (eq 7) being linear are much less clear. I accept that simple models often give more tractability and so more insight, and that this was enough to steer your experimental results, but I worry over the general applicability of the model. I'd certainly be interested to see what happens if you assume some of these

processes saturate.

Figures 1C and 1D are the key plots for your theoretical results. These two things have to be viewed in conjunction to understand the key result. I wonder if there is a better way of presenting these, or perhaps of adding some simulated dynamics in the SI. I couldn't completely understand what figure 1E is showing. Are these lines the assumptions of the model, or outputs from it?

On line 320 I think the subscript for the second theta is incorrect?

In equation (8), it is not clear where the numeric subscripts have suddenly come from.

Reviewer #3 (Remarks to the Author):

Costa et al report some interesting biological phenotypes for Plasmodium parasites but have presented the data in a way that attempts to address the evolution of virulence trajectories based on competition or parasitism states within the mosquito. This adds a layer of conjecture to the paper that I am unsure is supported by the data, although I am at the edge of my detailed knowledge here. Depletion of the lipid transporter Lipophorin (Lp) in female Anopheles mosquitoes caused a plethora of interesting parasite developmental defects within the mosquito. Transmission of the likely defective sporozoite progeny to mice resulted in less patent blood infections and less severe outcomes (cerebral malaria). The authors conclude that this represents virulence defects, but it could be that the parasites are sick/dying before they get into the mammalian host. This then is not necessarily a virulence defect but may be a developmental issue. Not all essential proteins are virulence factors. So, interesting as the results certainly are, different conclusions are quite possible in my opinion.

Major points:

To be honest, as a cell biologist I could not make much sense of Figure 1 as it was very field-specific. Can the descriptions be improved to reach a broader audience?

More assessment of true virulence mechanisms rather than lifecycle progression should be teased out. For example, is the gliding motility of Lp knockdown-derived sporozoites affected? Do they traverse and invade cells like Ctrl? Where do the parasites arrest first? Can they infect the liver normally? Is there a delay to patency? Knowing this would identify whether the defect is really stage-transcending at the erythrocyte, or whether it is the direct outcome of sick sporozoites.

Complementing with a second blood meal is elegant but again no virulence mechanisms have specifically been assessed. The restoration of in vitro EEF development suggests that liver stages are affected by Lp knockdown but direct evidence in mice is lacking. Is gliding motility affected following the second blood meal? Invasion frequency? Or is this an EEF developmental effect post-invasion?

Minor points

Figure 2: Were less parasites inoculated? In panel A, was the number of exposures to mosquitoes/bites equal? Did the mosquitoes have different salivary gland sporozoite numbers (means per experiment)? How many times was the experiment repeated (technical replicates)? Also, is there a liver stage defect first?

Figure 3. Panel C, how many mosquitoes? How many times was this experiment repeated? Panel F requires a P value even if it is non-significant, to prove so.

Figure S1: Please quantify the amount of Nile red labelling in ovaries and fat bodies to clearly show the effect from multiple images from multiple mosquitoes of Lp knockdown. The figure legend says "days 7 and 13" yet these are absent from the figure. The text (line 108) says parasite load is shown but I could not find this data in the figure.

Figure S2: In panel B, error bars are needed. The number of repeats should be $n=3$ here, and throughout the paper.

Figure S3: Why does the Ctrl oocyst in panel A (bottom left with sporozoites) look so different to the Ctrl oocyst in panel B (upper right with no sporozoites), when both are day 13?

Figure S4: Panel A, the CSP labelling looks less in dsLp than Ctrl. A single representation is probably not sufficient in the current era of rapid microscopy quantification. Some quantification of numerous sporozoites from different experimental replicates would judiciously address whether CSP, a major virulence factor, is involved. Panel D, should the axis title 'CCCP' be 'dsLp'?

Line 24: regarding 'between host and the mosquito vector'. Technically, both are hosts, with the mammal being an intermediate host and the insect being a definitive host.

Line 28: infective-to-vector forms?

Line 34: thousands OF sporozoites

Line 56: is 'evolution' the appropriate term?

Line 112: 'parasitizes mosquito lipids' should be re-worded. Sequesters?

Line 158: 'virulence-associated trait'. The current data may preclude such a strong conclusion.

Line 187: The overall conclusion regarding time-shift is unsupported. The data is interesting but is being over interpreted in my opinion.

Rebuttal Letter

We are grateful for the provided opportunity to submit our revised manuscript. We would like to thank Prof Sarah Reece and two anonymous reviewers for evaluating our manuscript and for their constructive comments and suggestions. We were delighted that all reviewers found our study interesting and timely. We agreed with most of the comments and tried to introduce new data or modify the original text to meet the reviewers' criticisms.

In the attempt to make it more accessible, we rewrote the modelling part, clearly stating the aims and major conclusions. This would not have been possible without a critical contribution of Dr Paola Carrillo-Bustamante. Therefore, we included her as a co-author. We also provide new data that identified parasite stages affected by Lp depletion and performed additional quantitative data on parasite viability and expression of the major sporozoite surface protein.

As one of the major criticisms related to the over interpretation of our results, we rewrote the manuscript (including the title) to separate the results from our interpretations. Note, that this editing does not change the major conclusions of the manuscript. All major modifications in the text are indicated by red color.

Please find below our point-by-point responses.

Reviewers' comments:

Reviewer #1

The manuscript tackles the long-neglected topic of within-vector ecology in a novel way, revealing unexpected and important biology. The authors are preaching to the converted when they contrast the depth of understanding of within-host dynamics and how selection has shaped parasite phenotypes expressed in the host to analogous information for the vector phase of the life cycle. Targeting parasite phenotypes in the vector offers novel intervention opportunities, but first, an understanding of such phenotypes in the context of both within- and between-host processes is required. The authors go a long way towards this by revealing a novel phenotype that could both block transmission and reduce virulence to humans.

Specifically, the finding that the within-mosquito environment modulates infectivity and virulence to the next host is the most striking result and I am sure it will have a significant impact on the field. Tracking the phenotype down to lipid availability is impressive too. Whilst on the whole I find the data to be compelling (I don't really have the expertise to comment on the technicalities of the model), I think their presentation could be made stronger and more accessible to a general audience. Thus, my comments at this stage are mostly fairly general.

1) The terminology and framework used in the model is rather confusing and only makes sense in light of the data. I would either rename the strategies or explain the logic of the nomenclature more clearly in the introductory text. The idea seems to be that mosquitoes get resources from blood that can be used for reproduction or used by the parasite. In a "competitive" scenario, the mosquito and parasite compete directly for resources as if they are free-living organisms competing over external resources. This view might be fine mathematically but is very confusing because the resources are actually owned by the mosquito; the mosquito is responsible for processing the blood into resources that are accessible to the parasite. Thus, this scenario is a form of parasitism and biologically hard to differentiate from the "parasitic scenario". I wonder if reading Clay Cressler's work (eg <http://onlinelibrary.wiley.com/doi/10.1111/ele.12229/epdf>) would be useful to help frame these scenarios more clearly?

We are grateful to the reviewer for pointing out that the model was neither sufficiently introduced nor described and apologize for this. The terminology "competition" and "parasitism" come from the classical Lotka-Volterra model, and we use it here considering that the resources acquired after a single blood meal to be a non-replenished nutrient source that is necessary for the development of mosquito eggs and parasite. In this theoretical sense, eggs and parasites engage in a direct symbiotic relationship, even if it is the mosquito that owns the resources. Specifically, eggs and parasites could be in direct competition for the nutrients (competitive scenario), or the parasites might exploit the dynamics of egg development to scavenge vector's resources (parasitic scenario). We believe that our theoretical approach, albeit abstract, provides a reasonably good indication of the biological processes in the vector, that cannot be directly tested experimentally. We then used experimental approaches to examine the key model prediction that the nature of the within-vector symbiotic relationships shapes the fitness of the parasite in a mammalian host.

Cressler's work is similar to our approach. It explores how both parasite and internal host processes use resources by modelling a resource that is being used by the parasite replication and fueling the host immune system. We could in principle use a similar framework, explicitly modelling a nutrient source and differentiating its allocation into a "parasite-priority" or "egg priority" model. However, such a modelling approach also requires a large number of parameter values for which we have insufficient knowledge. Given that we cannot use this theoretical framework, we feel it is more appropriate not to use the same terminology as Cressler, to avoid confusion.

We have extensively modified all parts related to the theoretical approach, and have included motivations for our theoretical model in the first part of the results (pages 4-6), clearly defining the two hypothetical scenarios. We also expanded the model description in the Material and Methods section (pages 15-18), and in the legend of the new Figure 1. We hope that with these additions, the theoretical approach is more accessible to a broad audience of readers.

- similarly, the last para of the discussion is quite confusing. It would help to be clear when virulence is mentioned, whether its to the host or vector.

We agree with the reviewer that this is a central point which needs a clear statement. In the first sentences of the introduction, we define parasite virulence "as the capacity to cause harm to the human host" (page 3, line 37-39). Every further use of this term in this manuscript is in line with this definition. We also modified the last paragraph of the Discussion to improve the clarity of our reasoning (page 13, lines 294-303).

2) Moreover, I am not entirely convinced the data can differentiate the model's scenarios and so I find it distracts me from the main strength of the study which is the very interesting data and implications. Perhaps either removing the model or presenting it after the data in the context of the results motivating theoretical exploration of the epidemiological consequences of a potential explanation for the delay in lipid uptake, will make the paper flow better and the strategies be easier to convey?

We thank the reviewer for raising this point. We believe that this confusion resulted from the insufficient description of our model. The overall aim of this study was to investigate how within-vector resource exploitation/acquisition affects within-host parasite virulence. The model generated here helped us framing the questions and designing the experiments. Importantly, it allowed us to develop theoretical predictions on how our experimental findings could influence the trajectories of parasite virulence. We thus believe the model is an essential part of this study and its position in the text serves our overall aim. To make our point clear, we have now modified all sections related to the model description (including results, figures and methods) to highlight the primary aim of the modelling part and to make it more accessible to a general audience (also see the answer to point #1).

3) It is clear that lipid depleted parasites are less infective, but think it might be premature to conclude their phenotype as less virulent in the next host. Even though the number of sporozoites is controlled for in some of the experiments, it could still be a function of infective dose. If a smaller proportion of sporozoites from lipid depleted vectors are healthy, then infections are effectively being started with fewer viable parasites. Is there evidence that even very very low numbers of "normal" sporozoites usually cause cerebral symptoms? This would add weight to the notion that something more than simply a reduction in the number of healthy sporozoites is happening.

*We thank the reviewer for this valuable comment, also raised by Reviewer 3 in her/his introductory paragraph. We agree that differences in sporozoite viability (infective dose) may cause lower infectivity and ECM. Although some studies reported that as little as 50 i.v. injected *P. berghei* sporozoites induce 100% patency in C57BL/6 mice (Scheller et al., 1994). To quantify sporozoite viability, we performed additional experiments to examine the viability of sporozoites by imaging flow cytometry and propidium iodide (PI) staining, a viability marker previously used for *Plasmodium* sporozoites (Purcell et al., 2008). We found that the proportion of live PI-negative sporozoites in lipid-deprived conditions is similar to controls (new Figure 4A) and corresponds on average to 95% of total sporozoites (as compared to 5% of PI-negative parasites heat-killed control). Moreover, GFP intensity (previously used as an alternative sporozoite viability marker - Yilmaz et al., 2014) is also identical in lipid-deprived and control sporozoites (Figure 4B). We conclude that the loss of virulence observed for lipid-deprived sporozoites does not result from a higher proportion of dead parasites.*

4) I have a few concerns about the analyses used, presumably due to the lack of detail it has been possible to include.

a- The sample sizes for mice in the experiments are not clear. I assume the n in Fig2 for A-C and D-F are number of mice monitored (it reads as if referring to numbers of mosquitoes dissected)?

These numbers represent the total number of mice monitored per experimental group. We apologize for the confusion and have modified the corresponding legend text to make this point clear (page 32, line 739 and 740-741).

b- Also, its not obvious what the number of mosquitoes per condition per experiment is. Eg Fig3 and 4 could be clearer in whether n refers to number of replicate experiments or mosquitoes within a condition. Just giving the number of oocysts is not helpful because mosquitoes or experiments are the independent replicates (see 4d).

We apologize for the confusion and have modified all figure legends to explicitly state the number of mosquitoes per condition per experiment, the number of experimental replicates (N) and, where applicable, the number of parasites analyzed. Moreover, we have introduced a table stating n per each experiment (Supplementary Table 2).

c- Being an evolutionary biologist, we are more picky about analyses than parasitology generally is. I'm worried about the use of parametric tests on n=3 samples. There are clear to see effects in the plots so it is unlikely type 1 errors are being made, but the using an anticonservative analysis may give false confidence in effect sizes.

We understand the reviewer's concern. All datasets were first tested for normality (Shapiro-Wilk Normality test) and homogeneity of variances (Levene's test) to choose the appropriate statistical method. In some cases, we performed additional experimental replicates to increase the sample size (new Fig. 5Bi and v and Fig. 3E). We have now added a sentence in the statistical analysis section of the Materials and Methods that clarifies this point (page 23, lines 513-515).

d- It looks like experiment means are presented in most figures but it is not clear whether data for individual oocysts, mosquito means, or experiment means are used in analyses. Whilst assaying lots of oocysts in each mosquito is better than assaying a few, it is not the case that each oocyst is an independent data point. Strictly speaking, each mosquito is a sample and the more oocysts examined, the more accurate the means are for each mosquito. In other words, oocysts sampled from the same mosquito are not independent of each other. The best way to analyse such data is to either "summarise" the data for each mosquito as its mean, and so groups of means can be compared with e.g. T tests or ANOVA. Or, with mixed effects models that use the data for each oocyst whilst accounting for the fact they belong to specific mosquitoes and different experimental blocks. If the authors have not used either of these approaches there could be issues of false confidence in effect sizes.

We thank the reviewer for pointing this out. We now have clarified in all figure legends where we show means of pooled data or means of single experiments. To compensate for a possible excess of confidence due to the inflated sample size of pooled data, we always used non-parametric tests (Mann-Whitney and Kruskal-Wallis tests).

e- Am I right in thinking the interaction plots and paired T-tests pair treatments within each experimental block? If so, clarifying this in the figure legends would be useful.

We thank the reviewer for pointing this out. Indeed, paired t-tests pair treatments were applied within each experimental block. The legend text has been modified accordingly (Figure 3B, and 4A-C, Figure 5B i and v, Supplementary Figure 2C, Supplementary Figure 5E-G).

5) I'm not convinced by conclusion that the ~1 week delay in lipid use by the parasite is explained as the parasite evolving to be restrained. I think the scenarios below merit consideration and discussion. As the authors state, determining the correct scenario matters for the evolutionary consequences of interventions that target lipid availability.

a- The idea that the mosquito is allowed to reproduce by the parasite and then allows its remaining lipids to be used by the parasite is not the conclusion I would come to first. It's not obvious to me that the parasite is restraining because the host could be in control and (somehow) not allowing lipids to be accessible to parasites for this time. Also, I'd assume the mosquito would have a use for these left-over lipids, for example, for subsequent bouts of reproduction. So, there could still be a "competitive relationship" but the vector is "winning".

Our interpretation, however speculative, considers two phases in Plasmodium sporogony: (i) pre-sporogonic, is the early phase after the ookinete rounding up in the basal labyrinth and (ii) sporogonic phase, that starts roughly one week after the blood feeding. We showed that only sporogonic parasites take up lipids. Indeed, lipid accumulation in the midgut during the first days after the first blood meal (Supplementary Figure 2A-B) did not benefit pre-sporogonic parasite development, suggesting that the ookinetes/early oocysts were incapable of lipid uptake. This observation is consistent with the first detection of Lp in the oocysts 7 days post infection (Supplementary Figure 3). As mosquito completes its gonotrophic cycle 2-3 days post blood feeding, we proposed that this shift in time prevents direct competition for mosquito lipids.

b- If it really is parasite restraint, then parasites should still show the delay in lipid extraction even in mosquitoes that have an excess of lipids (is it possible to give a blood meal with enhanced lipids, or infecting mosquitoes after their first egg lay when they have spare lipids?).

We thank the reviewer for this interesting suggestion. We agree this would be a way to experimentally investigate how evolutionary fixed/genetically regulated is the delayed lipid uptake in oocysts. However, this aspect is beyond of the scope of this manuscript (see also answer to point 5d) and will be addressed in future studies.

c- It could be a cost of crowding (this would still be very interesting): the availability of lipids determines the number of sporozoites that can successfully be matured. If so, it is not a parasite strategy but an unavoidable consequence of an environmental constraint. The authors say "Nutrient restriction in natural conditions could be induced by oocyst crowding; however in our experiments the numbers of pre-sporogonic oocysts between the experimental groups were similar." I don't see how these comments relate. The same numbers of oocysts mean that those in depleted mosquitoes are resource limited; ie fewer resources per capita.

We agree with the reviewer. By "oocysts crowding effect" we alluded to the scenario when sharing of resources by variable numbers of pre-sporogonic oocysts in the mosquito midgut can cause resource limitation (Pollitt et al., 2013). Lp silencing reduces ookinete numbers in wild-type mosquitoes but not in the immunocompromised mosquitoes (Rono et al., 2010). As we used immunocompromised mosquitoes, the number of pre-sporogonic oocysts was similar in the two experimental conditions. In this case, oocysts resource limitation can only be due to the experimentally impaired mosquito lipid trafficking and not to the differences in the initial numbers of invaded ookinetes. We have modified the sentence to clarify this concept (page 12, lines 255-258).

d- Furthermore, claiming a phenotype to be an adaptive strategy generally requires more support. Eg showing that parasites ultimately lose out on fitness if they use resources sooner. As an evolutionary biologist, one immediately considers a competitive scenario: if multiple parasite genotypes are present in the mosquito, the genotype that extracts lipids first does best (because it gets more lipids than its competitor) which results in selection for an arms race for earlier lipid extraction. Given that mosquitoes without lipid reserves don't seem to have short lifespans, there won't be a trade-off with premature death of the vector to constrain the timing of lipid extraction. Also, the parasites interests would be best served by scavenging lipids for the purpose of stopping the mosquito from reproducing. It has become clear that infection reduces mosquito lifespan but only when mosquitoes reproduce. Thus, if the parasite were able to extract lipids earlier they might mature a greater proportion of healthy sporozoites and extend the lifespan of the vector.

We thank the reviewer for this comment. We agree that showing that a phenotype is an adaptive strategy requires evolutionary experiments, which we have not attempted in this study. In Figure 5 we provide evidence that in normal conditions the uptake of lipids by oocysts starts only around 7 dpi. Importantly, Lp depletion results in lipid accumulation in the midgut and aborts ovary development. In the case of competitive scenario, one would expect that these conditions should benefit parasite's fitness and increase within-host virulence. Instead, we observed the opposite outcome. Therefore, we proposed that the oocyst competence for lipid uptake regulates a time-shift between mosquito reproduction and Plasmodium sporogony. We have modified the corresponding text in Results page 10, lines 208-214: "Lp depletion caused lipid accumulation in the midguts of blood-fed mosquitoes (Supplementary Figure 2). However, this lipid-rich environment after the blood meal did not benefit sporozoite fitness and attenuated the within-host infectivity and virulence even in the absence of direct competition with egg development. As only sporogonic oocysts accumulated neutral lipids (Figure 3A), we hypothesized that early oocysts were incompetent in lipid uptake, instead, providing lipids at the initiation of oocyst sporogony should rescue parasite development and virulence."; and page 10, 229-230: "We concluded that oocysts acquire lipids after completion of vector's oogenesis." We also changed the corresponding part in Discussion: page13, lines 294-303: "Our data further suggest that a time-shift between vector oogenesis and parasite sporogony rules out resource competition scenario between vector and parasite. Indeed, the parasites develop competence for lipid uptake after the completion of egg development one week after infection. This profitable developmental niche is retained as long as nutrient supply is sufficient. Changes in the nutrient environment should similarly impact oogenesis and parasite sporogony, explaining the positive correlation between ovary development and parasite loads often observed in experimental infections. Further investigation of mosquito and parasite factors involved nutrient acquisition and metabolism is necessary to predict potential switches between parasitic and competitive vector-parasite relationships."

e- Having said the above, there is an evolutionary scenario (Mafia hypothesis, Restif & colleagues) that could apply. In this scenario, the host gives the parasite resources in return for restraining its virulence.

While it is an interesting idea, it does not apply here because pre-sporogonic parasites are incapable of competing with the vector (see above).

6) Are there data on egg output? It would be nice to show that infected control mosquitoes do actually reproduce. If they don't, then the hypothesis that the parasite is restraining to allow reproduction of the vector is not supported.

Approximately 80% of infected control mosquitoes developed mature eggs. Data on egg development are shown in Supplementary Figure 2C.

7) I don't believe the data shows the evolution part of the statement that "that evolution of malaria virulence is

metabolically regulated by the within-vector nutritional environment". Within-vector resources are a mechanism that regulates virulence. Demonstrating the "evolution" part would require revealing eg signatures of selection on genes involved in resource extraction or a change in these processes if within-host virulence is experimentally perturbed in an artificial selection regime. This is just an example of precision in evolutionary terminology and does not detract from the value of the results.

See also answer to Reviewer 3, minor points comment to line 56. We agree with the reviewer and modified the sentence accordingly. Page 4, lines 73-77: "By combining theoretical predictions and experimental data, we propose that the within-vector nutritional environment regulates malaria virulence. This finding has important potential implications for vector control strategies based on biological competitors or manipulation of mosquito reproduction."

8) It would increase the impact for a general audience if the results for malaria infection could be sat in the context of other diseases showing similar resource use strategies? Eg is "parasite restraint" a common strategy in nature or are Plasmodium seemingly novel in this regard? Including such info in the intro would provide valuable context for why the impact of within-vector processes on within-host processes is investigated.

It is hard to argue for a common parasitic strategy as the parasites interact with diverse hosts, exhibit intracellular and extracellular life cycles and a multitude of invasion strategies. We didn't find any evidence in this direction in the literature.

9) A final thought is Spence et al Nature showed mosquito transmission per se attenuates virulence, probably via eliciting different immune responses to blood passaged parasites. Is it possible that in normal circumstances*, some lipid deprived parasites will always occur in mosquitoes and could be responsible for eliciting the phenotype Spence et al observed? (*eg species that do not replicate to such high numbers in mosquitoes)

We thank the reviewer for this valuable comment. Spence et al. demonstrated that vector passage intrinsically modifies the asexual blood-stage parasites and the elicited mammalian immune response and associated malaria pathology. In this study, attenuated parasite virulence correlated with changes in expression of the P. chabaudi multi-gene pir family during mosquito passage. To explore possible links between this phenotype and parasite lipid deprivation within the mosquitoes, we screened the expression level of several bir genes (orthologous to P. chabaudi pirs) expressed in sporozoites (BIR_0400500, BIR_0722900, BIR_0500971, BIR_0836900). The expression levels of all genes measured by qRT-PCR were below detection levels except for BIR_0722900 (data not shown). However, we did not observe any significant differences in expression levels of BIR_0722900 between control and lipid-deprived sporozoites (see figure below). A further transcriptomic analysis is required to explore expression of other variable genes at the genome level.

Sarah Reece

Reviewer #2

The manuscript presents both theoretical and experimental results exploring how within-vector resource competition impacts malaria virulence. The theoretical work shows that if there is straight competition between the vector and parasite for resources that there will be runaway virulence, but that if the relationship is truly parasitic then virulence can be stabilised. They then show experimentally that the parasite may be avoiding competition by delaying its replication.

I should say at the outset that I am a theoretician, and that I therefore do not feel fully confident assessing the experimental procedures. I thought the manuscript examined an important question, the methodologies (both theory and experiment) appear well planned, they find a novel result and the manuscript itself is well-written. The combination of theory and experiment is a real strength, and identifying how malaria parasites may be avoiding resource competition is interesting both theoretically but also from a treatment perspective. I have focussed my comments below on the theoretical model.

My only real question is to do with the fact that so many of the modelled processes are linear. That vector-to-human transmission (eq 5) is linear in P is fairly common (though vector-borne diseases are often modelled as being frequency-dependent), but the justification for virulence (eq 6) and competition (eq 7) being linear are much less clear. I accept that simple models often give more tractability and so more insight, and that this was enough to steer your experimental results, but I worry over the general applicability of the model. I'd certainly be interested to see what happens if you assume some of these processes saturate.

We thank the reviewer for raising this important point. We have also developed a model in which virulence saturates depending on the initial parasite load, i.e., $v_h(\bar{P}) = v_0 \alpha_2 \bar{P} / (h + \bar{P})$, where h represents the parasite density at which the virulence is half maximal (see Material and Methods, page 18, lines 380). As shown in a new Supplementary Figure 7, including non-linear responses in our epidemiological model only changes the total number of R_0 (200 instead of 100) but does not qualitatively affect our results. We still observe that the optimal fitness of Plasmodium is achieved only if we assume parasitic interactions and for intermediate virulence values.

The within-vector Lotka-Volterra models are implicitly saturating, as we always consider density dependent terms (i.e., the carrying capacities).

Former eq 7 refers to our assumption of virulence being a function of the within-vector interaction strength α_2 , considering that the parasite's capacity of resource exploitation in the vector will also determine its replication ability in the mammalian host. Adding here a saturating process would not qualitatively change our results, as it would be effectively the same change as explained above.

Figures 1C and 1D are the key plots for your theoretical results. These two things have to be viewed in conjunction to understand the key result. I wonder if there is a better way of presenting these, or perhaps of adding some simulated dynamics in the SI. I couldn't completely understand what figure 1E is showing. Are these lines the assumptions of the model, or outputs from it?

We thank the reviewer for raising these concerns. We realize that the Figure 1 and the overall model description were somewhat confusing. We re-designed Figure 1 by including a better model cartoon with our key results (indeed, former panel E was showing model assumptions, and we have removed it). In particular, we split heatmaps into parasitic and competitive scenarios, introduced white arrows to highlight the areas of interest and amended the legend to reflect these changes. We sincerely hope that these efforts make our take-home message clear.

Note that we did not explicitly model the infection dynamics here but only calculated R_0 . Therefore, we cannot show any SI dynamics. We hope that the new explanations in the main text (pages 4-6) and Material and Methods (pages 15-18), as well as the new figures make this part more accessible to a broader audience.

On line 320 I think the subscript for the second theta is incorrect?

See below.

In equation (8), it is not clear where the numeric subscripts have suddenly come from.

We thank the reviewer for pointing out these two points. We corrected these small typos in the equations and re-wrote the entire section (pages 15-18).

Reviewer #3

Costa et al report some interesting biological phenotypes for Plasmodium parasites but have presented the data in a way that attempts to address the evolution of virulence trajectories based on competition or parasitism states within the mosquito. This adds a layer of conjecture to the paper that I am unsure is supported by the data, although I am at the edge of my detailed knowledge here. Depletion of the lipid transporter Lipophorin (Lp) in female Anopheles mosquitoes caused a plethora of interesting parasite developmental defects within the mosquito. Transmission of the likely defective sporozoite progeny to mice resulted in less patent blood infections and less severe outcomes (cerebral malaria).

The authors conclude that this represents virulence defects, but it could be that the parasites are sick/dying before they get into the mammalian host. This then is not necessarily a virulence defect but may be a developmental issue. Not all essential proteins are virulence factors. So, interesting as the results certainly are, different conclusions are quite possible in my opinion.

We thank the reviewer for this valuable comment, also raised by Reviewer 1. We defined parasite virulence in the Introduction "as the capacity to cause harm to the human host" (page 3, lines 41-43). We acknowledge the classical microbiologist's view brought up by the reviewer, which defines virulence factors as components of a pathogen that, when deleted, impair virulence but not viability (Casadevall and Pirofski, 1999). To quantify sporozoite viability, we performed additional experiments using imaging flow cytometry and propidium iodide (PI) staining, a viability marker previously used for Plasmodium sporozoites (Purcell et al., 2008). We found that the proportion of live PI-negative sporozoites in lipid-deprived conditions is similar to controls (new Figure 4A) and corresponds to 95% of total population (for comparison, heat-killing treatment results in \approx 95% of dead parasites). Moreover, GFP intensity (previously used as an alternative sporozoite viability marker (Yilmaz et al., 2014) is also identical in lipid-deprived and controls sporozoites (Figure 4B). We conclude that the loss of virulence observed for lipid-deprived sporozoites is not caused by a higher proportion of dead parasites.

Major points:

To be honest, as a cell biologist I could not make much sense of Figure 1 as it was very field-specific. Can the descriptions be improved to reach a broader audience?

We thank the reviewer for this comment. We revised the model description, highlighted the primary aim of the model and believe that now this theoretical part is more accessible to a broader audience. Also, see our answer to Reviewer 1 (point 2) and Reviewer 2 (comment to Fig. 1C and 1D).

More assessment of true virulence mechanisms rather than lifecycle progression should be teased out. For example, is the gliding motility of Lp knockdown-derived sporozoites affected? Do they traverse and invade cells like Ctrl? Where do the parasites arrest first? Can they infect the liver normally? Is there a delay to patency? Knowing this would identify whether the defect is really stage-transcending at the erythrocyte, or whether it is the direct outcome of sick sporozoites.

The major conclusion of our study is that mosquito lipids shape Plasmodium virulence as manifested by a delay to patency and lower proportion of mice displaying symptoms of experimental cerebral malaria (shown in Figure 2). This phenotype is observed across mosquito stages starting from oocyst development and culminating at the number of developing EEFs in the hepatocytes, gauged by in vitro developmental assays (shown in Figures 3-5). Therefore, we describe a complex phenotype that goes beyond a single critical step but concerns multiple stages, each of which contributes to within-host infectivity and virulence. We provide additional data on the delay to patency of lipid-deprived parasites as compared to controls in the new Figure 2F. The parasites isolated from lipid-deprived mosquitoes recover normal growth after infecting the host red blood cells as illustrated by parallel slopes of parasitemia curves. We find these data surprising because of the striking impact of the within-vector environment that cannot be rescued by a parasite sojourn in a lipid-rich liver. The stage-transcending phenotype is observed in oocyst-sporozoite- EEF transition but not in the blood stages. Please see page 7, lines 146-148.

Complementing with a second blood meal is elegant but again no virulence mechanisms have specifically been assessed. The restoration of in vitro EEF development suggests that liver stages are affected by Lp knockdown but direct evidence in mice is lacking. Is gliding motility affected following the second blood meal? Invasion frequency? Or is this an EEF developmental effect post-invasion?

As discussed above and shown in Figure 5B, a second blood meal restores the number sporozoites produced by mosquitoes, the numbers and sizes of EEFs 48h post seeding. Because of the time point chosen for this experiments, we cannot discriminate between sporozoite traversal/invasion defects or post-invasion EEFs death. We now provide new data showing that the supplemental blood feeding restores sporozoite virulence in vivo (Supplementary Figure 6B-D). Consistent with our in vitro results, a supplemental blood feeding abolished the observed differences in prepatency period and the incidence of severe disease in mice infected by subcutaneous injections of 5,000 sporozoites isolated from control and Lp-depleted mosquitoes. We noted an overall lower ECM incidence in these experiments (Supplementary Figure 6C) compared to the ones performed two/three years

earlier (Figure 2E). This difference in mice susceptibility could result from the change of the animal provider in May 2017 (Villarino et al., 2016). For transparency, this information has been added into the Materials and Methods (page 21, lines 456-458): "C57BL/6J mice were purchased from Charles River until 2016 (Figure 2) or bred in the internal institute core facility from 2017 (Supplementary Figure 6).".

Minor points

Figure 2: Were less parasites inoculated? In panel A, was the number of exposures to mosquitoes/bites equal? Did the mosquitoes have different salivary gland sporozoite numbers (means per experiment)? How many times was the experiment repeated (technical replicates)? Also, is there a liver stage defect first?

The number of mice and experiments presented in Figure 2 are now explicitly stated in the figure legend and in Supplementary Table 2. For panel A and B, mice were exposed to the same number of mosquitoes per group (ranging from 13 to 18 per experiment, N=4). We added this information to the Materials and Methods section page 21, (lines 462-463). Panels C-F display disease progression in mice injected with the equal number of sporozoites (5,000 per mice, N=3). The data on sporozoite numbers in Ctrl and dsLp mosquitoes is shown in Figure 3E.

As discussed above, the newly added panels demonstrate that the delay in prepatency and lower numbers of ECM cases in bite-back experiments are caused by lower sporozoite quantities (Figure 2A-B and Figure 3E) and by lower sporozoite infectivity/liver development (Figure 2C-F). As soon as the parasites invade the red blood cells, the parasite amplification rates are similar between the two groups (new Figure 2F, see also comments above).

Figure 3. Panel C, how many mosquitoes? How many times was this experiment repeated? Panel F requires a P value even if it is non-significant, to prove so.

We thank the reviewer for pointing this out, see also answer to Reviewer 1. The number of mosquitoes per experiment is now clearly stated in each panel and detailed in Supplementary Table 2. Figure 5C shows TEM analysis of oocyst (5-14 mosquitoes per condition per experiment, N=1 (7 dpi) and N=3 (13 dpi)). The results are shown for one representative experiment which was chosen because of the highest numbers of oocysts. This information has been now added in the corresponding figure legend (page 33, lines 747-749). The p-value was added in the figure legend of former panel F, current Figure 4B (page 34, line 774).

Figure S1: Please quantify the amount of Nile red labelling in ovaries and fat bodies to clearly show the effect from multiple images from multiple mosquitoes of Lp knockdown. The figure legend says "days 7 and 13" yet these are absent from the figure. The text (line 108) says parasite load is shown but I could not find this data in the figure.

We thank the reviewer for these valuable suggestions. We quantified mosquito lipids in the relevant tissues by Nile Red staining and added the information on the numbers of examined mosquitoes (Supplementary Figure 2B, page 2, lines 18-22). "Days 7 and 13" were removed from the legend of panel A. In the text (previous line 108) "parasite load" was removed.

Figure S2: In panel B, error bars are needed. The number of repeats should be n=3 here, and throughout the paper.

This experiment was performed two times with very consistent results. We now provide information on the number of examined oocysts per time point per experiment (Supplementary file, page 4, lines 43-44).

Figure S3: Why does the Ctrl oocyst in panel A (bottom left with sporozoites) look so different to the Ctrl oocyst in panel B (upper right with no sporozoites), when both are day 13?

P. berghei oocysts undergo asynchronous development. We defined oocysts' morphological classes at 13 dpi and plotted quantitative estimates for each class in Figure 3C (half of the oocysts show variable stages of sporulation, one quarter is immature and one quarter is abnormal/dead). New supplementary Figures 4A and B (former supplementary Figure 3) reflect the observed variability in oocysts' development.

Figure S4: Panel A, the CSP labelling looks less in dsLp than Ctrl. A single representation is probably not sufficient in the current era of rapid microscopy quantification. Some quantification of numerous sporozoites from different experimental replicates would judiciously address whether CSP, a major virulence factor, is involved. Panel D, should the axis title 'CCCCP' be 'dsLp'?

We agree with the reviewer and performed three additional experiments using imaging flow cytometry to quantify CSP protein expression in control and lipid-deprived sporozoites using anti-CSP antibody (new Supplementary Figure 5E and F). These additional data are consistent with the CSP transcript quantification by qRT-PCR and show no differences in CSP expression between Ctrl and dsLp sporozoites.

Panel G (previously D) shows control experiments that validate TMRE as a reporter dye of the sporozoite mitochondrial activity. In these experiments, sporozoites isolated from control mosquitoes were treated with the chemical inhibitor of oxidative phosphorylation carbonyl cyanide m-chlorophenyl hydrazone (CCCP). Our data confirm the specificity of the TMRE signal, which is undetectable in the CCCP treated sporozoites. This information is now stated in the main text (page 9, lines 190-192): "We first confirmed dye specificity by examining TMRE fluorescence after sporozoite treatment with the inhibitor of oxidative phosphorylation CCCP (Supplementary Figure 5G).".

Line 24: regarding 'between host and the mosquito vector'. Technically, both are hosts, with the mammal being an intermediate host and the insect being a definitive host.

We agree with this classical definition and introduced a complete description in the Introduction (page 3, lines 41-43) but use the shorter host and vector nomenclature through the text.

Line 28: infective-to-vector forms?

Thank you, the text was changed accordingly (page 3, line 47).

Line 34: thousands OF sporozoites

Thank you, the text was changed accordingly (page 3, line 55).

Line 56: is 'evolution' the appropriate term?

Please see also comment to Reviewer 1. We agree with the reviewer comment and modified this sentence accordingly (page 4, lines 73-77): "By combining theoretical predictions and experimental data, we propose that the within-vector nutritional environment regulates malaria virulence. This finding has important potential implications for vector control strategies based on biological competitors or manipulation of mosquito reproduction.".

Line 112: 'parasitizes mosquito lipids' should be re-worded. Sequesters?

We changed the title accordingly (page 8, lines 156-158): "As restricting lipid transport attenuated within-host virulence even in the absence of ovary development, we considered the parasitic scenario, where Plasmodium exploits mosquito Lp for lipid delivery.".

Line 158: 'virulence-associated trait'. The current data may preclude such a strong conclusion.

To avoid confusion between the results and our interpretations, we changed this sentence. Page 9, lines 201-202: "Further, we show that within-vector lipid environment shapes mitochondrial membrane potential of Plasmodium transmissible stages.".

Line 187: The overall conclusion regarding time-shift is unsupported. The data is interesting but is being over interpreted in my opinion.

Please see also answer to Reviewer 1. We apologize for over-interpreting the data and have modified the corresponding text. In Figure 5 we provide evidence that in normal conditions the uptake of lipids by oocysts starts only around 7 dpi. Importantly, Lp depletion results in lipid accumulation in the midgut and aborts ovary development. In the case of competitive scenario, one would expect that these conditions should benefit parasite's fitness and increase within-host virulence. Instead, we observed the opposite outcome. Therefore, we proposed that the oocyst competence for lipid uptake regulates a time-shift between mosquito reproduction and Plasmodium sporogony. We have modified the corresponding text in Results page 10, lines 208-214: "Lp depletion caused lipid accumulation in the midguts of blood-fed mosquitoes (Supplementary Figure 2). However, this lipid-rich environment after the blood meal did not benefit sporozoite fitness and attenuated the within-host infectivity and virulence even in the absence of direct competition with egg development. As only sporogonic oocysts accumulated neutral lipids (Figure 3A), we hypothesized that early oocysts were incompetent in lipid uptake, instead, providing lipids at the initiation of oocyst sporogony should rescue parasite development and virulence."; and page 10-11, 227-230: "We concluded that oocysts acquire lipids after completion of vector's oogenesis." We also changed the corresponding part in Discussion: pages 13-14, lines 294-303: "Our data further suggest that a time-shift between vector oogenesis and parasite sporogony rules out resource competition scenario between vector and parasite. Indeed, the parasites develop competence for lipid uptake after the completion of egg development one week after infection. This profitable developmental niche is retained as long as nutrient supply is sufficient. Changes in the nutrient environment should similarly impact oogenesis and parasite sporogony, explaining the positive correlation between ovary development and parasite loads often observed in experimental infections. Further investigation of mosquito and parasite factors involved nutrient acquisition and metabolism is necessary to predict potential switches between parasitic and competitive vector-parasite relationships."

REVIEWERS' COMMENTS:

Reviewer #1 (Remarks to the Author):

The authors have done an excellent job at clarifying many aspects of the paper. The model is now well explained and integrated and the new data make the case for the importance of vector lipids to parasites even more compelling. Most of my comments are aimed at further clarification but in some areas, I am still a little nervous about making causative connections between the model and data.

(1) in response to my comment that:

The idea that the mosquito is allowed to reproduce by the parasite and then allows its remaining lipids to be used by the parasite is not the conclusion I would come to first. It's not obvious to me that the parasite is restraining because the host could be in control and (somehow) not allowing lipids to be accessible to parasites for this time. Also, I'd assume the mosquito would have a use for these left-over lipids, for example, for subsequent bouts of reproduction. So, there could still be a "competitive relationship" but the vector is "winning".

Our interpretation, however speculative, considers two phases in Plasmodium sporogony:, (i) pre-sporogonic, is the early phase after the ookinete rounding up in the basal labyrinth and (ii) sporogonic phase, that starts roughly one week after the blood feeding. We showed that only sporogonic parasites take up lipids. Indeed, lipid accumulation in the midgut during the first days after the first blood meal (Supplementary Figure 2A-B) did not benefit pre-sporogonic parasite development, suggesting that the ookinetes/early oocysts were incapable of lipid uptake. This observation is consistent with the first detection of Lp in the oocysts 7 days post infection (Supplementary Figure 3). As mosquito completes its gonotrophic cycle 2-3 days post blood feeding, we proposed that this shift in time prevents direct competition for mosquito lipids.

Agreed, but this delay could still be explained by the vector having a way to prevent lipids being used by others (eg other parasites/ microbes etc). If so, the parasites might have to had to evolve a strategy not to use lipids in this period and this is why it cannot take them up – because it is costly to express a trait before it will bring any fitness returns. I think my scenario is unlikely but it is a remote possibility and in light of the next comments, perhaps present the time-shift as consistent with the parasitic strategy rather than as a demonstration of it?

(2) Line 236/244/304: I am still a little nervous about the conclusion that the "that oocysts acquire lipids after completion of vector's oogenesis". Because there aren't data on lipid uptake in the parasite before day 7 it is impossible to rule out unexpected lipid dynamics in parasites at days 2-6. To be fair, the data do suggest lipid uptake is likely negligible before day 7 but without examining this, it shouldn't be assumed. Make this conclusion a little more tentative?

(3) The dynamics of lipid uptake by the parasite raise another important question: If the

parasite is simply waiting until the mosquito has used lipids for oogenesis – which takes 2 days – then why does the parasite delay taking up lipids until ~day 7? If Lp / lipids are the most important factor and they are available to the parasite from day 3 PI, then why doesn't sporogony happen faster? Addressing this question with some (speculation) on other limitations (eg temperature? other resources) governing rates of sporogony would be useful. I wouldn't suggest this if the aim of the paper was simply to show how lipids matter to parasites, but because the aim is to evaluate lipid use in the context of a parasite virulence strategy it is important to consider other phenomena impact on this phenotype and so, the evolution of a strategy. The worry is that parasites are delaying until day 7 to start sporogony due to something else.

minor comments

Title: specify that its virulence to the mammalian host to highlight the novelty of the findings

Line 90: The use of "competition" and "parasitism" is now clearer but does require some thinking about. To avoid readers not used to these concepts can you give some examples here to differentiate them further? eg under competition, intuitively, the parasite should be selected to garner all the resources and prevent the mosquito from using any for its own reproduction. Under parasitism, the resources are more readily extracted by the parasite when the mosquito undergoes oogenesis, so the parasite requires the mosquito to undergo some reproduction. I know this info comes at the end of the paragraph but my feeling is that detailed the scenarios first will help orient non specialists and then the paragraph can simply be concluded by info that these predictions are supported by the model.

Line 95: define alpha rather than assume readers know what this parameter is

Line 135: perhaps mention something at the start of this section about why the focus is on lipids rather than other types of resource used by the parasite (eg sugars)?

Line 151: is there a "not" missing?

Line 159: to make the case very clear, is it worth pointing out more clearly that: lipids were present in the Lp mosquitoes but the parasites is unable to acquire them, thus the parasite likely requires Lp for lipid delivery. If so, the parasite requires a mosquito transporter involved in oogenesis for its resource acquisition, thus supporting the parasitic scenario.

Line 204: Add the conclusion that the results demonstrate a trade-off between sporozoite quality and quantity, mediated by lipid availability?

Line 736: perhaps state that N refers to the number of experiments over which n infections were monitored per treatment

Sarah Reece

Reviewer #2 (Remarks to the Author):

I previously reviewed this manuscript, making clear that I felt mainly qualified to comment on the theoretical model. I am pleased to see the authors have acted on my comments and I have only one concrete concern to add.

The very confusing figure 1 has been replaced with something much more intuitive, which is great.

I'm also pleased they checked what would happen if virulence saturated. I had wondered if limit cycles might appear, and while it doesn't look to me like you explicitly checked that, after some checking of the literature I suspect that it is unlikely anyway.

One thing does jump out at me in the revision is that you make repeated comments about 'evolutionary trajectories' (L84, L127, L160, L397). Trajectory implies some explicit *dynamic* evolutionary modelling, but you just look for where R_0 is maximised, a static condition. The white arrows in figure 1E (and figure S1) don't help with this as they give the impression of evolution moving in a certain direction. I suspect everything you say would still be true - the model is still simple enough that I would not expect any complicated feedbacks to arise that prevent the parasite maximising R_0 (which can happen in a number of models), but I still think the word trajectory is misleading.

Reviewer #3 (Remarks to the Author):

The authors have now addressed my concerns. I share excitement with the authors about what the manuscript now shows in revision. Congratulations. I recommend acceptance after the below minor suggestions are considered.

Line 727: "...occurring at intermediate virulence values.". This looks to be "low" virulence values to me. See also the author's own wording "low" on line 119.

Figure 1. Having panels D and E in the model was challenging to decode, although I persisted and eventually understood it. Only when I considered the model as a single panel with the following text (line 725) did it become clear to me:

"The white arrows (left panels) point to the contours with the highest parasite virulence values (R_0) and to the corresponding values (right panels) of virulence in the human host (V_h)." The authors could consider altering the text in such a way for clarity, but it is up to them.

Figure 2. The legend for panel B has "***" but there are no asterisks on the graph. Which variables were compared? Were statistics performed for panel D?

Supplementary Figure 4. The authors state in the rebuttal that they plotted quantitative estimates for each oocyst class in panel C. However, there is no panel C (nor was there in the prior version of the manuscript).

Supplementary Figure 5. Are the data in panels E and F the “mean”? I understand that 3 experimental replicates are shown in these panels, yet the error within each experiment is not shown. Is this standard?

Supplementary Figure 6. Have statistics been performed on the data in panels A and D?

REVIEWERS' COMMENTS:

Reviewer #1 (Remarks to the Author):

The authors have done an excellent job at clarifying many aspects of the paper. The model is now well explained and integrated and the new data make the case for the importance of vector lipids to parasites even more compelling. Most of my comments are aimed at further clarification but in some areas, I am still a little nervous about making causative connections between the model and data.

(1) in response to my comment that:

The idea that the mosquito is allowed to reproduce by the parasite and then allows its remaining lipids to be used by the parasite is not the conclusion I would come to first. It's not obvious to me that the parasite is restraining because the host could be in control and (somehow) not allowing lipids to be accessible to parasites for this time. Also, I'd assume the mosquito would have a use for these left-over lipids, for example, for subsequent bouts of reproduction. So, there could still be a "competitive relationship" but the vector is "winning".

Our interpretation, however speculative, considers two phases in Plasmodium sporogony: (i) pre-sporogonic, is the early phase after the ookinete rounding up in the basal labyrinth and (ii) sporogonic phase, that starts roughly one week after the blood feeding. We showed that only sporogonic parasites take up lipids. Indeed, lipid accumulation in the midgut during the first days after the first blood meal (Supplementary Figure 2A-B) did not benefit pre-sporogonic parasite development, suggesting that the ookinetes/early oocysts were incapable of lipid uptake. This observation is consistent with the first detection of Lp in the oocysts 7 days post infection (Supplementary Figure 3). As mosquito completes its gonotrophic cycle 2-3 days post blood feeding, we proposed that this shift in time prevents direct competition for mosquito lipids.

Agreed, but this delay could still be explained by the vector having a way to prevent lipids being used by others (eg other parasites/ microbes etc). If so, the parasites might have to evolve a strategy not to use lipids in this period and this is why it cannot take them up – because it is costly to express a trait before it will bring any fitness returns. I think my scenario is unlikely but it is a remote possibility and in light of the next comments, perhaps present the time-shift as consistent with the parasitic strategy rather than as a demonstration of it?

We agree with the reviewer and changed our discussion accordingly: Page 15, lines 319-322: "Our hypothesis of a time-shift between vector oogenesis and parasite sporogony is consistent with the parasitic scenario. Indeed, the parasites may develop competence for lipid uptake after the completion of egg development one week after infection."

(2) Line 236/244/304: I am still a little nervous about the conclusion that the "that oocysts acquire lipids after completion of vector's oogenesis". Because there aren't data on lipid uptake in the parasite before day 7 it is impossible to rule out unexpected lipid dynamics in parasites at days 2-6. To be fair, the data do suggest lipid uptake is likely negligible before day 7 but without examining this, it shouldn't be assumed. Make this conclusion a little more tentative?

We agree with the reviewer and rephrased our conclusion: Page 12 lines 247-249: "Taken together these results are in line with our hypothesis that oocysts acquire mosquito lipids after completion of vector's oogenesis."

(3) The dynamics of lipid uptake by the parasite raise another important question: If the parasite is simply waiting until the mosquito has used lipids for oogenesis – which takes 2 days – then why does the parasite delay taking up lipids until ~day 7? If Lp / lipids are the most important factor and they are available to the parasite from day 3 PI, then why doesn't sporogony happen faster? Addressing this question with some (speculation) on other limitations (eg temperature? other resources) governing rates of sporogony would be useful. I wouldn't suggest this if the aim of the paper was simply to show how lipids matter to parasites, but because the aim is to evaluate lipid use in the context of a parasite virulence strategy it is important to consider other phenomena impact on this phenotype and so, the evolution of a strategy. The worry is that parasites are delaying until day 7 to start sporogony due to something else.

Indeed, our study proposes but does not answer the question why the parasites wait for 5 additional days before acquisition of mosquito lipids. We think that this question has never been addressed, reflecting the paucity of studies on mosquito stages of Plasmodium. To make our message more clear, we introduced the following discussion on page 13 lines 269-274: "Currently, the processes that regulate parasite's competence to acquire lipids remain unknown. Perhaps, the extracellular location of the oocysts imposes some restrictions on the rate of parasite sporogony as opposed to intracellular sporogony of red blood stages. Future comparative studies of transcriptional profiles of young and sporogonic oocysts may reveal new players in this process, whose manipulation will be essential to confirm or refute our hypothesis."

Minor comments

Title: specify that its virulence to the mammalian host to highlight the novelty of the findings

We thank the reviewer for this comment and changed the title accordingly: "Non-competitive resource exploitation within-mosquito shapes within-host malaria infectivity and virulence".

Line 90: The use of "competition" and "parasitism" is now clearer but does require some thinking about. To avoid readers not used to these concepts can you give some examples here to differentiate them further? eg under competition, intuitively, the parasite should be selected to garner all the resources and prevent the mosquito from using any for its own reproduction. Under parasitism, the resources are more readily extracted by the parasite when the mosquito undergoes oogenesis, so the parasite requires the mosquito to undergo some reproduction. I know this info comes at the end of the paragraph but my feeling is that detailed the scenarios first will help orient non specialists and then the paragraph can simply be concluded by info that these predictions are supported by the model.

We thank the reviewer for her comment. We have included the suggested examples in the same paragraph (page 5 lines 95-98): "Under competition, the parasite would benefit most if it acquired all resources, thereby preventing the mosquito from using any for its own reproduction. Under parasitism, the parasite would garner the resources initiated by the vector's oogenesis, thus not only benefiting from, but requiring egg development."

Line 95: define alpha rather than assume readers know what this parameter is

We have included the explanation in the same brackets.

Line 135: perhaps mention something at the start of this section about why the focus is on lipids rather than other types of resource used by the parasite (eg sugars)?

We thank the reviewer for this suggestion and changed the text accordingly on page 8 lines 145-147: "To formalize our theoretical assumption, we evaluated whether restricting within-vector resources impacted within-host Plasmodium virulence and focused our studies on lipids that have been previously implicated in within-vector parasite development¹⁴⁻¹⁶".

Line 151: is there a "not" missing?

If the reviewer means this part of the sentence: "As the observed phenotypes could result from the differences in numbers of inoculated parasites,..", "not" is not missing here. But to avoid confusion, we changed this sentence: "However, the observed difference in the number of infected mice and ECM cases may result from unequal numbers of inoculated parasites."

Line 159: to make the case very clear, is it worth pointing out more clearly that: lipids were present in the Lp mosquitoes but the parasites is unable to acquire them, thus the parasite likely requires Lp for lipid delivery. If so, the parasite requires a mosquito transporter involved in oogenesis for its resource acquisition, thus supporting the parasitic scenario.

We thank the reviewer for this excellent suggestion and introduced the following conclusion on page 9 lines 165-169: "Despite lipid accumulation in the midgut after a blood meal, depletion of Lp attenuated within-host infectivity and virulence of P. berghei parasites. These results suggest that the within-mosquito parasite development likely relies on the lipid transporter involved in oogenesis, and thus endorse the parasitic scenario."

Line 204: Add the conclusion that the results demonstrate a trade-off between sporozoite quality and quantity, mediated by lipid availability?

We apologise for the confusion. Our results suggest a positive correlation between sporozoite quantity and quality. We modified our conclusion on page 11 line 218-220: "Further, we show that within-vector lipid environment shapes mitochondrial membrane potential and the quantity and the quality of Plasmodium transmissible stages."

Line 736: perhaps state that N refers to the number of experiments over which n infections were monitored per treatment

We changed the legend to Figure 2 accordingly.

Sarah Reece

Reviewer #2 (Remarks to the Author):

I previously reviewed this manuscript, making clear that I felt mainly qualified to comment on the theoretical model. I am pleased to see the authors have acted on my comments and I have only one concrete concern to add.

The very confusing figure 1 has been replaced with something much more intuitive, which is great.

I'm also pleased they checked what would happen if virulence saturated. I had wondered if limit cycles might appear, and while it doesn't look to me like you explicitly checked that, after some checking of the literature I suspect that it is unlikely anyway.

We thank the reviewer for this comment. As mentioned in the earlier revision, we do not explicitly model the infection dynamics, but only calculate R_0 for different $\alpha_{1/2}$ values. Therefore, we could not check for any instability behavior of the infection. In the within-vector model, limit cycles could occur but not in the parameter regime used in our model.

One thing does jump out at me in the revision is that you make repeated comments about 'evolutionary trajectories' (L84, L127, L160, L397). Trajectory implies some explicit *dynamic* evolutionary modelling, but you just look for where R_0 is maximised, a static condition. The white arrows in figure 1E (and figure S1) don't help with this as they give the impression of evolution moving in a certain direction. I suspect everything you say would still be true - the model is still simple enough that I would not expect any complicated feedbacks to arise that prevent the parasite maximising R_0 (which can happen in a number of models), but I still think the word trajectory is misleading.

We fully agree in that the term 'trajectory' might be misleading as we indeed study the evolution of virulence under static conditions. To avoid confusion, we have removed it from the entire text.

Reviewer #3 (Remarks to the Author):

The authors have now addressed my concerns. I share excitement with the authors about what the manuscript now shows in revision. Congratulations. I recommend acceptance after the below minor suggestions are considered.

Line 727: "...occurring at intermediate virulence values.". This looks to be "low" virulence values to me. See also the author's own wording "low" on line 119.

We thank the reviewer for this comment and changed the wording accordingly on page 32 line 731.

Figure 1. Having panels D and E in the model was challenging to decode, although I persisted and eventually understood it. Only when I considered the model as a single panel with the following text (line 725) did it become clear to me:

"The white arrows (left panels) point to the contours with the highest parasite virulence values (R_0) and to the corresponding values (right panels) of virulence in the human host (V_h)." The authors could consider altering the text in such a way for clarity, but it is up to them.

We thank the reviewer for this excellent suggestion. We have merged both panels into one and corrected the legend and main text accordingly.

Figure 2. The legend for panel B has "****" but there are no asterisks on the graph. Which variables were compared? Were statistics performed for panel D?

*We apologise for this ambiguity and changed the Figure 2 legend, which now reads: "C57BL/6J mice bitten by *P. berghei*-infected mosquitoes (A-B) or infected by sporozoite injection (C-F) were daily monitored for parasite occurrence in the blood and for symptoms of experimental cerebral malaria (ECM). (A) Kaplan-Meier analysis of time to infection (blood stage parasitemia) of mice infected by bites of control (Ctrl) or Lp-depleted (dsLp) mosquitoes (N - number of experiments: N=4, n - total number of mice per each group: n=10). **: p<0.001; log-rank test. (B) Cumulative health status of mice shown in A. (C) Kaplan-Meier analysis of time to infection (blood stage parasitemia), (D) incidence (%) of experimental cerebral malaria (ECM) and (E) parasitemia (mean \pm SEM) in mice infected by subcutaneous injection of 5,000 sporozoites dissected from the salivary glands of control (Ctrl) or Lp-depleted (dsLp) mosquitoes (N=3, total number of mice in Ctrl and dsLp groups, n=18 and n=16, respectively). Asterisks indicate statistically significant differences (*: p<0.05; **: p<0.001; ***: p<0.0001 log-rank test (C and D) and two-sided Mann-Whitney test comparing Ctrl and dsLp conditions for each time point (E)). (F) Cumulative health status of mice shown in C, E and F."*

Please note that no additional statistical analyses have been performed on the pooled data.

Supplementary Figure 4. The authors state in the rebuttal that they plotted quantitative estimates for each oocyst class in panel C. However, there is no panel C (nor was there in the prior version of the manuscript).

We are sorry for the confusion, we referred to panel C in Figure 3, not in Supplementary Figure 4.

Supplementary Figure 5. Are the data in panels E and F the “mean”? I understand that 3 experimental replicates are shown in these panels, yet the error within each experiment is not shown. Is this standard?

The dots in the former panel E show the percentage (proportions) of CSP-positive sporozoites per experiment, while the dots in the former panel F shows the geometric mean fluorescence intensities (MFI) per experiment. To show sporozoite variability within a single experiment, we have introduced a new panel (E) with fluorescence intensity histograms of one representative experiment. We also updated the Figure legend which now reads: “Fluorescence intensity histograms from one representative experiment (E), proportions of CSP-positive sporozoites (F) and CSP geometric MFI (G) of each experiment are shown.”

Supplementary Figure 6. Have statistics been performed on the data in panels A and D?

As Supplementary Figure 6A summarizes the data shown in Figure 5E and I, and Supplementary Figure 6D summarizes the data shown in panel B and C, all information on statistical analyses is presented in the original figures and no additional statistical analyses have been performed on pooled data.